# MULTI-ORDER WAVELET DERIVATIVE TRANSFORM FOR DEEP TIME SERIES FORECASTING

## ABSTRACT

In deep time series forecasting, the Fourier Transform (FT) is extensively employed for frequency representation learning. However, it often struggles in capturing multi-scale, time-sensitive patterns. Although the Wavelet Transform (WT) can capture these patterns through frequency decomposition, its coefficients are insensitive to abrupt changes in the time series, leading to suboptimal modeling. To mitigate these limitations, we introduce the multi-order Wavelet Derivative Transform (WDT) grounded in the WT, enabling the extraction of time-aware patterns spanning both the overall trend and subtle fluctuations. Compared with the standard FT and WT, which model the raw series, WDT operates on the derivative of the series, selectively magnifying rate-of-change cues and exposing abrupt regime shifts that are particularly informative for time series modeling. Practically, we embed the WDT into a multi-branch framework named **WaveTS**, which decomposes the input series into multi-scale time-frequency coefficients, refines them via linear layers, and reconstructs them into the time domain via the inverse WDT. Extensive experiments on multiple benchmark datasets demonstrate that WaveTS achieves state-of-the-art forecasting accuracy while retaining high computational efficiency. The code is available at https://anonymous.4open.science/r/WaveTS-4review.

## 1 INTRODUCTION

Time series forecasting aims to predict future values by leveraging patterns of past observations, playing a pivotal role in diverse fields such as climate, finance, and transportation (Wang et al., 2020; Zhang et al., 2023; Wang et al., 2024c; Yang et al., 2025; Qiu et al., 2025; Kieu et al., 2019). In recent years, deep learning techniques have gained prominence for modeling long-term temporal dependencies, reflected in MLP-based (Zeng et al., 2023; Zhang et al., 2022), RNN-based (Jia et al., 2023; Hewamalage et al., 2021), Transformer-based (Liu et al., 2024c; Nie et al., 2023; Wu et al., 2024) etc., underscoring the effectiveness of temporal representation learning in the **time domain**.

Alongside these time-domain methods, recent deep forecasting models have also been proposed to model time series in the **frequency domain**, namely **F**requency domain **R**epresentation **L**earning (**FRL**). These FRL-based forecasters largely leverage the Fourier transform for domain transformation, thereby extracting the frequency components. Subsequently, they process these frequency components using complex-valued neural networks to derive semantic frequency representations. For example, FreTS (Yi et al., 2023b) introduces frequency domain MLPs to effectively model both intra-variate and inter-variate dependencies. FITS (Xu et al., 2024) incorporates frequency interpolation through a complex-valued linear layer, rendering it 50 times more compact than previous linear models. Additionally, several studies have incorporated the FRL paradigm into more complex architectures like Transformers and Graph Neural Networks to extract more semantic and robust features in the frequency domain (Zhou et al., 2022b; Eldele et al., 2024; Cao et al., 2020; Yi et al., 2023a).

However, previous FRL-based forecasters exhibit a heavy reliance on the Fourier transform for domain transformation (Yi et al., 2023b; Xu et al., 2024; Wang et al., 2025c; Yi et al., 2024a; Fan et al., 2014; Fei et al., 2025). While this methodology is both effective and computationally efficient, it is inherently constrained in its ability to model time-sensitive patterns. As illustrated in Figure 1 (a)-(d), similar frequency spectra are generated after the application of the Fourier transform to time series exhibiting markedly distinct temporal variations. This leads to ambiguities in deep representation learning, as well as inadequate modeling of the time-frequency characteristics and non-stationarity of time series. DeRiTS (Fan et al., 2014) attempts to address this issue by proposing a deep Fourier

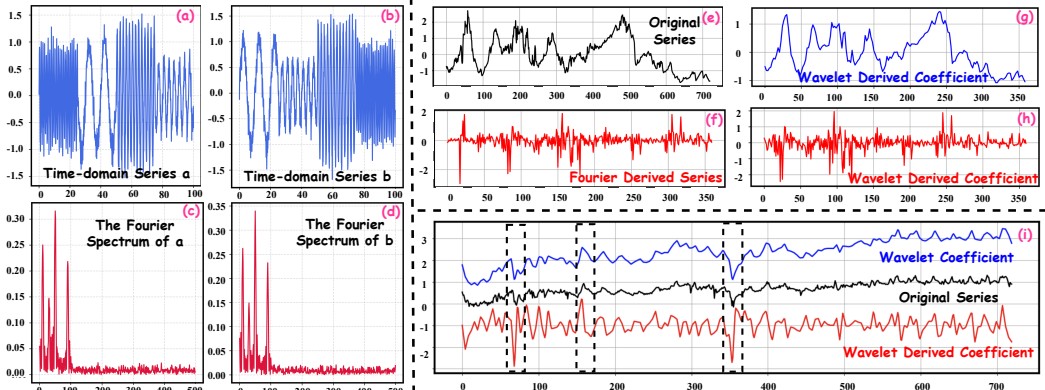

Figure 1: (a)–(d) Distinct temporal variations may yield identical frequency spectra after applying the Fourier transform, revealing spectral ambiguity; (e) original time series; (f) the Fourier Derived Series only exhibits high-frequency fluctuations with values around 0 and fails to capture macro temporal patterns such as the overall trend of the original series in (e); (g) the Wavelet Derivative Transform extracts macro low-frequency trend patterns; (h) the Wavelet Derivative Transform extracts micro high-frequency fluctuations; (i) relative to the wavelet coefficients in the standard wavelet transform (blue curve), WDT coefficients (red curve) pinpoint abrupt regime shifts, yielding a sharper cue for non-stationary dynamics.

derivative learning framework aimed at learning stationary frequency representations (Figure 1 (f)). However, this approach neglects to consider macro temporal patterns, such as the overall trend, and still fails to capture time-aware features, ultimately resulting in inaccurate forecasting.

To address the aforementioned issues, this paper proposes a new transformation for time series, namely Wavelet Derivative Transform (WDT). WDT is built upon the Wavelet Transform (WT), targeted in modeling the derivative of time series rather than the raw series. It decomposes time series into slowly varying trend coefficients and stationary oscillatory coefficients, revealing hierarchical (macro) and stationary (detail) patterns (Figure 1 (g) and (h)).

Moreover, as shown in Figure 1 (i), the coefficient of standard WT (blue curve) predominantly smooths the signal and simply fits its overall trend, ignoring the depiction of abrupt change points in black dotted boxes. However, by encoding the series' derivative, the coefficient of WDT (red curve) not only preserves the macroscopic trend, but also highlights local bursts more sharply than the standard wavelet coefficient, providing richer cues for pinpointing sharp discontinuities, thereby facilitating its perception of local dynamics. Based on WDT, we propose a multi-branch architecture **WaveTS**, where each branch corresponds to a WaveTS block designated to process a specific order of WDT. WDT first decomposes time series into time-frequency coefficients. Subsequently, Frequency Refinement Units, comprising multiple hardware-friendly real-valued linear layers, refine these coefficients into more granular time-frequency representations, enabling WaveTS to extract richer patterns across various spectral bands. Ultimately, the inverse WDT reconstructs both historical and future sequences back into the time domain. To make this multi-order construction mathematically sound, we rigorously analyze WDT and prove that it defines a well-posed linear transform with an exact inverse and stable multi-order coefficients. These results clarify when WDT preserves the energy and information content of the original series and explain why its derivative-aware coefficients can be safely exploited in downstream models. Our contributions can be articulated as follows:

- We propose a multi-order Wavelet Derivative Transform that captures multi-scale time–frequency patterns, enabling sharper localization of regime shifts than the standard wavelet transform.

- We provide a rigorous theoretical analysis of the proposed Wavelet Derivative Transform, proving the existence of an exact inverse transform and establishing energy-preserving properties, thereby clarifying the mathematical difficulty and significance of our construction.

- Built upon our proposed Wavelet Derivative Transform (with its inverse), we introduce a new multi-branch deep forecasting model, namely WaveTS, for effective multivariate time series forecasting.

- WaveTS unequivocally outperforms previous state-of-the-art methods, achieving $5.1\%$ MSE reduction with the lowest memory usage and fastest inference speed, demonstrating its efficiency.

## 2 PRELIMINARIES

### 2.1 WAVELET TRANSFORM

The Discrete Wavelet Transform (DWT) Mallat (1989) decomposes a discrete time series signal into time-frequency components. Given a time series $[X(1), \ldots, X(N)]$ of length $N$, the DWT coefficient $W_{k,j}$ at scale $k$ and translation $j$ is given by:

$$W_{k,j} = \sum_{n=0}^{N-1} X(n)\,\psi_{k,j}[n], \ k = 0, \ldots, K-1, j = 0, \ldots, \tfrac{N}{2^k} - 1, \tag{1}$$

where $\psi_{k,j}[n] = 2^{-k/2}\psi\left[2^{-k}n - j\right]$ is the discrete basis function with mother wavelet $\psi[n]$, and $K = \lfloor \log_2 N \rfloor$ is the maximum dyadic scale. At each scale $k$, the signal is split into a high-frequency (detail) component capturing the abrupt changes, and a low-frequency (approximation) component reflecting the overall trends. Multi-resolution analysis of both the time and frequency components is obtained by repeatedly decomposing the low-frequency components. The Haar (Haar, 1911) and Daubechies (Daubechies, 1988) wavelets are commonly used. They provide orthogonal or nearly orthogonal decompositions, ensuring that the DWT coefficients at different scales and translations capture non-redundant features of the signal (Liu et al., 2024a).

### 2.2 RELATED WORK

**Deep Learning for Time Series Forecasting**  Deep learning-based methodologies have been extensively applied to time series forecasting, as evidenced by TCNs (Wu et al., 2023; Liu et al., 2022), and Transformer-based models (Liu et al., 2024c; Nie et al., 2023; Wang et al., 2024b; Zhou et al., 2021). More recently, LLM-based methods (Zhou et al., 2023; Jin et al., 2024; Liang et al., 2024) have emerged, leveraging the capacity of large pre-trained language models for knowledge transfer. Alongside these, FRL-based approaches (Yi et al., 2023b; Xu et al., 2024; Yi et al., 2023a; 2024a;b; Zhou et al., 2022a; 2026) employ the Fourier transform to uncover frequency patterns implicit in time-domain signals. FAN (Ye et al., 2024) identifies non-stationary factors by extracting dominant real frequency components, while DeRiTS (Fan et al., 2014) represents time series through frequency derivation. However, existing FRL-based models still face challenges in capturing the inherently multi-scale time-frequency features, leading to ineffective time series modeling.

**Wavelet Transform for Time Series Analysis**  In contrast to Fourier-based transformations, the wavelet transform decomposes signals into localized time-frequency components, making it well-suited for multi-scale analysis (Zhou et al., 2022b; Stefenon et al., 2023; Gao et al., 2021; Ouyang et al., 2021). For example, wMDN (Wang et al., 2018) embeds wavelet-based frequency analysis into a deep learning framework through fine-tuning. AdaWaveNet (Yu et al., 2024) employs an adaptive and learnable wavelet decomposition scheme to address non-stationary data, while T-WaveNet (Liu et al., 2021) adopts a tree-structured network for iteratively disentangling dominant frequency subbands. WPMixer (Murad et al., 2025) couples multi-resolution wavelet decomposition with patching and MLP mixing. WaveletMixer (Zhang et al., 2025) is an iterative multi-level, multi-resolution, and multi-phase design that captures long- and short-term dependencies. However, most wavelet-coefficient pipelines do not explicitly address regime shifts and thus perform worse around transitions, even if overall accuracy is competitive. Augmenting them with shift-sensitive mechanisms (e.g., derivative-enhanced transforms) improves alignment near abrupt changes.

## 3 METHODOLOGY

Though standard wavelet-based forecasters can provide a hierarchical scale-by-scale decomposition, their coefficients mainly encode amplitude and thus under-emphasize multi-scale rates of change. This makes them less effective at highlighting local regime shifts and at modeling non-stationary temporal patterns or time-varying distribution shift within a time series (i.e., the change of its mean, variance over time). In this paper, we propose the Wavelet Derivative Transform (WDT) in Section 3.1 with its inverse, which explicitly models the derivatives of a time series to sharpen multi-scale change signals. Building on it, we further propose **WaveTS**, a multi-branch architecture built upon WaveTS blocks (Section 3.2.2) that applies multi-order WDT, which linearly refines the resulting coefficients through Frequency Refinement Unit, and reconstructs them back to the time domain for accurate and efficient backcasting while forecasting (Section 3.2.3).

In the following, we denote a multivariate time series by $[X_1, X_2, \ldots, X_T] \in \mathbb{R}^{T \times C}$, where each $X_t \in \mathbb{R}^C$ corresponds to the observations of $C$ variates at the $t$-th time step. We use a backcasting and

forecasting approach to align the model forecasts with the observed historical trend (Cao et al., 2020). Specifically, at time $t$, the model input $\mathbf{X}_t = [X_{t-L+1}, X_{t-L+2}, \ldots, X_t] \in \mathbb{R}^{L \times C}$ is a window of $L$ observations, and the model output is $\mathbf{Y}_t = [\hat{X}_{t-L+1}, \ldots, \hat{X}_{t-1}, \hat{X}_t, \hat{X}_{t+1}, \ldots, \hat{X}_{t+\tau}] \in \mathbb{R}^{(L+\tau) \times C}$, where $[\hat{X}_{t-L+1}, \ldots, \hat{X}_{t-1}, \hat{X}_t]$ are the backcasting results and $[\hat{X}_{t+1}, \ldots, \hat{X}_{t+\tau}]$ are the forecasting results. The model, parameterized by $\theta$, denoted $f_\theta(\cdot)$, utilizes the historical data $\mathbf{X}_t$ to estimate the backcasting and forecasting values $\mathbf{Y}_t = f_\theta(\mathbf{X}_t)$.

## 3.1 WAVELET DERIVATIVE TRANSFORM (WDT)

While the classical WT decomposes a series across scales, applying it directly to the raw values mostly captures magnitude information and does not explicitly single out rapid trend jumps, key symptoms of non-stationarity. In this section, we propose the Wavelet Derivative Transform (WDT), which models the derivative of time series. This disentangles the non-stationary trends and stationary variations in the time-frequency domain, leading to more effective modeling.

However, computing the derivative directly in the time domain is fragile: finite derivatives amplify high-frequency noise, shorten the usable sequence and complicate boundary handling. WDT avoids these pitfalls by performing derivative of the wavelet basis instead of the raw series—a spectral-domain trick that keeps the transform perfectly invertible, suppresses numerical noise and preserves scale alignment. Put simply, **the WT of the multi-order derivative of $X(t)$, can be expressed as the WT of $X(t)$ using the multi-order derivative of the mother wavelet**. Therefore, WDT captures derivative information without the usual time-domain baggage. See detailed proof in Appendix B.1.

**Definition 1** (Wavelet Derivative Transform (WDT)). *Let $X(t)$ ($t = 0, \ldots, T-1$) be a discrete time series, the WDT is the wavelet transform of $X(t)$ using the $n$-order derivative of the mother wavelet $\psi^{(n)}(t)$ as:*

$$\text{WDT}_{k,j}^{(n)} = (-1)^n \, 2^{nk} \, \tilde{W}_{k,j}^{(n)}.$$

*Here, $\tilde{W}_{k,j}^{(n)} = \sum_{t=0}^{T-1} X(t) \, \psi_{k,j}^{(n)}(t)$ is the wavelet transform of $X(t)$ using the $n$-order derivative of the mother wavelet $\psi^{(n)}(t)$, where $\psi_k^{(n)}(t)$ is the $n$-order derivative of the discrete wavelet basis function, defined as: $\psi_{k,j}^{(n)}(t) = 2^{k/2} \psi^{(n)}(2^k t - j)$.*

**Definition 2** (Inverse Wavelet Derivative Transform (iWDT)). *To reconstruct $X(t)$ from its wavelet derivative coefficients, the inverse wavelet transform is applied with the appropriate scaling factors, resulting in inverse Wavelet Derivative Transform (iWDT):*

$$X(t) = \text{iWDT}_K^{(n)}\big(\mathcal{D}^{(n)}(W_{k,j})\big) = \sum_{k=1}^{K} \sum_j \frac{\mathcal{D}^{(n)}(W_{k,j})}{(-1)^n \, 2^{nk}} \, \psi_{k,j}(t) = \sum_{k=1}^{K} \sum_j \tilde{W}_{k,j}^{(n)} \, \psi_{k,j}(t), \quad (2)$$

*where $\mathcal{D}^{(n)}(W_{k,j})$ denotes the $n$-th-order WDT coefficient at scale $k$ (there are total $K$ scales) and translation $j$, $\psi_{k,j}(t) = 2^{-k/2}\psi(2^{-k}t - j)$ is the standard discrete wavelet basis associated with the mother wavelet $\psi$, and $\tilde{W}_{k,j}^{(n)} = (-1)^{-n} 2^{-nk} \mathcal{D}^{(n)}(W_{k,j})$ is the rescaled coefficient that matches a conventional inverse WT. Because the rescaling is exact and the inverse WT is perfectly reconstructive, iWDT restores $X(t)$ without information loss; a Parseval-type proof of its energy-conservation property is provided in Appendix B.2.*

## 3.2 WAVETS

Figure 2 shows the proposed WaveTS, which takes sequence $\mathbf{X}_t$ as input and outputs sequence $\mathbf{Y}_t$. $\mathbf{X}_t$ is first normalized (Section 3.2.1) before feeding into $N$ parallel branches (WaveTS blocks). Each block corresponds to a specific order of the Wavelet Derivative Transform (WDT) and its inverse, facilitating the capture of granular time-frequency patterns through Frequency Refinement Units. Subsequently, the representations obtained from the blocks are concatenated along the temporal dimension. They are then projected and denormalized to obtain the output sequence $\mathbf{Y}_t$. WaveTS is the first model to tackle the limitations mentioned in Section 1 in FRL-based methods (including both Fourier-based and Wavelet-based methods) by integrating the WDT into a multi-branch architecture, significantly improving forecasting accuracy.

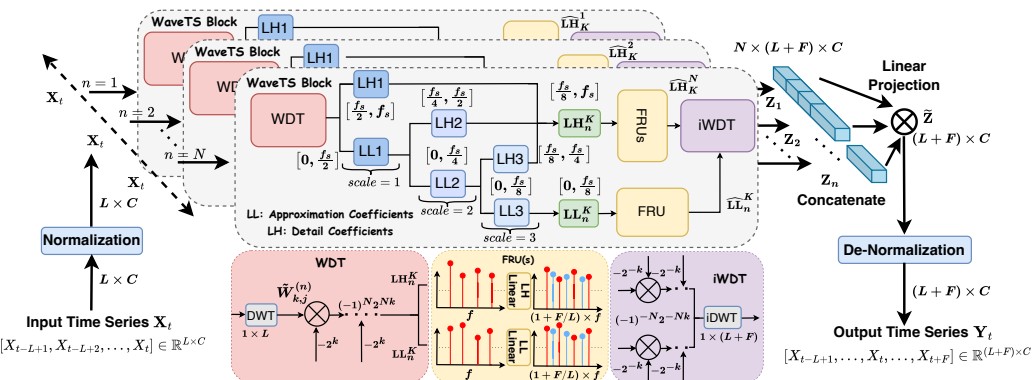

Figure 2: The main architecture of our proposed WaveTS. For each branch where $n$ starts from 1 to $N$, first-order to $N$-th order of Wavelet Derivative Transforms (WDT) are applied to capture multi-scale representations, facilitating more effective time series modeling. LH and LL are frequency coefficients that capture representations at different scales, corresponding to micro and macro patterns, respectively. These coefficients are transformed back to the time domain through Inverse Wavelet Derivative Transforms (iWDT). Finally, the representations from all branches are concatenated along the time dimension and projected to obtain the final forecast.

### 3.2.1 Instance Normalization

Following many previous works (Liu et al., 2024c; Xu et al., 2024; Yi et al., 2024a), WaveTS uses instance normalization (Kim et al., 2021), facilitating more accurate forecasting. For the input window $\mathbf{X}_t = [X_{t-L+1}, X_{t-L+2}, \ldots, X_t] \in \mathbb{R}^{L \times C}$, instance normalization produces: $\text{Norm}(\mathbf{X}_t) = \left[ \frac{X_{c,t} - \text{Mean}_L(X_{c,t}^{1:L})}{\text{Std}_L(X_{c,t}^{1:L})} \right]_{c=1}^{C}$, where $\text{Mean}_L(X_{c,t}^{1:L})$ is the mean value along the time dimension, and $\text{Std}_L(X_{c,t}^{1:L})$ is the corresponding standard deviation. Here, $X_{c,t}^{1:L}$ refers to the first $L$ observations of the $c$-th variate in $\mathbf{X}_t$.

### 3.2.2 WaveTS Block

As shown in Figure 2, our proposed WaveTS is built upon a multi-branch architecture, in which each branch constitutes a WaveTS block dedicated to a specific order of the originally proposed Wavelet Derivative Transform (WDT). Within each WaveTS Block, the input time series is decomposed using the WDT to derive frequency coefficients. Subsequently, Frequency Refinement Units (FRUs) process these coefficients to yield fine-grained frequency representations. Finally, the inverse WDT (iWDT) is employed to convert the learned representations back into the time domain.

**Time Series Decomposition with WDT** Based on the proposed WDT, WaveTS consists of $N$ branches, each utilizing a WT of the total scale $K$, as shown in Figure 2. For the $n$-th branch ($n = 1, 2, \ldots, N$), after instance normalization, we apply an $n$-order WDT ($\text{WDT}_{K,j}^{(n)}$) to the input sequence $\mathbf{X}_t$:

$$\left\{ \mathbf{LH}_K^{(n)}, \ LL_K^{(n)} \right\} = \text{WDT}_{K,j}^{(n)}(\mathbf{X}_t), \tag{3}$$

where $LL_K^{(n)} \in \mathbb{R}^{\frac{L}{2^K} \times C}$ is a matrix containing the low-frequency (macro) coefficient and $\mathbf{LH}_K^{(n)} = \{LH_1^{(n)}, \ldots, LH_K^{(n)}\}$ contains multiple matrices of the $K$ high-frequency (detail) coefficients at scale $K$. Specifically, for each branch, $\text{WDT}_{K,j}^{(n)}$ performs total $K$ levels of decomposition. At each decomposition level $k \in [1, K]$, the transform decomposes the current approximation coefficient $LL_{k-1}^{(n)}$ into a new approximation coefficient $LL_k^{(n)}$ and a detail coefficient $LH_k^{(n)}$. This hierarchical, tree-structured decomposition captures multi-scale information, ensuring that the final approximation coefficient $LL_K^{(n)}$ corresponds to the frequency band $\left[0, \frac{f_s}{2^K}\right]$, and the final detail coefficient $\mathbf{LH}_K^{(n)}$ covers the frequency band $\left[\frac{f_s}{2^K}, f_s\right]$. Consequently, the full frequency spectrum $[0, f_s]$ is comprehensively utilized. Note that the standard WT processes univariate input series rather than multivariate input, the input time series $\mathbf{X}_t$ is processed in a channel-independent way in each WaveTS block.

**Frequency Refinement Unit (FRU)**   After WDT-based decomposition, each branch $n$ produces one low-frequency coefficient $LL_K^{(n)}$ and multiple high-frequency coefficients $\mathbf{LH}_K^{(n)} = \{LH_1^{(n)}, \ldots, LH_K^{(n)}\}$. The Frequency Refinement Units comprise multiple linear layers, subsequently refine these coefficients by expanding their length from $L$ to $L + F$ through frequency interpolation as introduced by FITS (Xu et al., 2024). Specifically, we define a resolution factor $\alpha$ as: $\alpha = \frac{L+F}{L}$, so that each length-$L$ coefficient vector is projected to $\alpha L = L + F$ discrete points. Formally,

$$\widehat{LL}_K^{(n)} = \text{FRU}\big(LL_K^{(n)}\big), \qquad \widehat{\mathbf{LH}}_K^{(n)} = \text{FRUs}\big(\mathbf{LH}_K^{(n)}\big). \tag{4}$$

When $K = 3$, the $\text{FRU}(\cdot)$ denotes a single linear layer to model $LL_K^{(n)}$ and the $\text{FRUs}(\cdot)$ comprises 3 distinct linear layers that do not share weights to model $\mathbf{LH}_K^{(n)}$. Therefore, by learning these mappings for low- and high-frequency coefficients separately, the $\text{FRU}(\cdot)$ effectively performs time-frequency coefficients super-resolution in a discrete format, yielding a richer spectral representation without altering the overall domain. The resulting higher-resolution frequency features can then be merged and passed to the iWDT seamlessly.

**Branch Aggregation**   After mapping each branch's detail and approximate coefficients (cf. Eq. 4), WaveTS applies the iWDT on each branch $n \in [1, N]$ to recover a time-domain representation $\mathbf{Z}_n$:

$$\mathbf{Z}_n = \text{iWDT}_K^{(n)}\big(\widehat{LL}_K^{(n)}, \widehat{\mathbf{LH}}_K^{(n)}\big) \in \mathbb{R}^{(L+F)\times C}. \tag{5}$$

We then concatenate all $N$ branches along the time dimension: $\widetilde{\mathbf{Z}} = \text{Concat}\big(\mathbf{Z}_1, \mathbf{Z}_2, \ldots, \mathbf{Z}_N\big) \in \mathbb{R}^{N\times(L+F)\times C}$, thus integrating the representations from multiple branches. Finally, a linear layer projects the time dimension from $N \times (L + F)$ to $L + F$, yielding the final output $\mathbf{Y}_t$:

$$\mathbf{Y}_t = \text{Projection}\big(\widetilde{\mathbf{Z}}\big) \in \mathbb{R}^{(L+F)\times C}. \tag{6}$$

This design fuses multi-order wavelet derivative coefficients into one sequence while preserving the desired dimension of $C$. Finally, the output sequence $\mathbf{Y}_t$ is obtained using inverse instance normalization: $\text{DeNorm}(\mathbf{Y}_t) = \big[Y_{c,t} \times \text{Std}_L(X_{c,t}^{1:L}) + \text{Mean}_L(X_{c,t}^{1:L})\big]_1^C$, where $\mathbf{Y}_t = [Y_{c,t-L}, \ldots, Y_{c,t+F}] \in \mathbb{R}^{(L+F)\times C}$, $\text{Std}_L(X_{c,t}^{1:L})$ and $\text{Mean}_L(X_{c,t}^{1:L})$ originate from the input $\mathbf{X}_t$.

### 3.2.3 TRAINING OBJECTIVE

Previously, the Direct Forecast (DF) paradigm has been widely adopted in deep time series forecasting (Liu et al., 2024c; Wu et al., 2023; Zhou et al., 2024; Huang et al., 2023). In DF, a multi-output model $f_\theta$ generates predictions for the next $F$ steps in one pass: $\hat{Y} = f_\theta(X)$, where $X$ is the input. The L2 loss is computed by summing over all horizons: $\mathcal{L}_{DF} = \sum_{f=1}^F \|\hat{Y}_f - Y_f\|^2$, where $Y_f$ is the ground truth (Wang et al., 2025a). As mentioned in Section 1, the frequency features in FRL are usually processed by neural networks (we illustrate the FRL paradigm for better understanding in Appendix A). However, in the simplest instantiation, e.g., a single complex-valued linear layer as in FITS (Xu et al., 2024) or a single circular convolution layer as in PaiFilter (Yi et al., 2024a) and DeRiTS (Fan et al., 2014), this mapping simplifies to a linear transformation, so the new spectrum is merely a linear combination of the original. Consequently, simply reusing the previously adopted DF supervision paradigm is problematic, since **it neglects the inherent presence of historical information within the future forecasting window in the frequency domain**, contradicting the core principle that frequency transforms naturally preserve past information. Simply put, the output spectrum already carries a "free copy" of the recent past spectrum, because each frequency component is only re-scaled and phased-shifted during the linear mapping. If training focuses solely on the future window, that readily available signal is wasted, leading to suboptimal time series modeling.

Based on the aforementioned analysis, WaveTS adopts a **backcasting while forecasting** supervision method in training. Adding a backcasting term converts the preserved past into an extra set of labels at no additional data cost, forcing the network to reproduce what it has implicitly retained. This self-supervised consistency narrows the solution space, speeds convergence, and anchors the learned spectrum, reducing the drift that appears when non-stationary series are modeled in the frequency domain (Xu et al., 2024; Cao et al., 2020; Liu et al., 2024b). Specifically, at each time $t$, WaveTS produces an extended output $\hat{\mathbf{Z}}_t \in \mathbb{R}^{(L+\tau)\times C}$, where the first $L$ steps backcast the ground-truth past

series $\mathbf{X}_t \in \mathbb{R}^{L\times C}$ and the last $\tau$ steps forecast ground-truth $\mathbf{Y}_t \in \mathbb{R}^{\tau\times C}$. By concatenating $\mathbf{X}_t$ and $\mathbf{Y}_t$ along the time dimension to form $[\mathbf{X}_t \oplus \mathbf{Y}_t] \in \mathbb{R}^{(L+\tau)\times C}$, the overall L2 loss $\mathcal{L}$ is formulated as:

$$\mathcal{L} = \underbrace{\frac{1}{CL}\left\| \hat{\mathbf{Z}}_t^{\text{history}} - \mathbf{X}_t \right\|_F^2}_{\text{backcast loss}} + \underbrace{\frac{1}{C\tau}\left\| \hat{\mathbf{Z}}_t^{\text{future}} - \mathbf{Y}_t \right\|_F^2}_{\text{forecast loss}} = \frac{1}{C(L+\tau)}\left\| \hat{\mathbf{Z}}_t - [\mathbf{X}_t \oplus \mathbf{Y}_t] \right\|_F^2. \quad (7)$$

This approach effectively unifies backcasting and forecasting within a single objective, ensuring that the model aligns future predictions with $\mathbf{Y}_t$ while simultaneously reconstructing the past $\mathbf{X}_t$.

# 4 Experiments

## 4.1 Experimental Setup

**Datasets** This paper conducts extensive experiments on 8 datasets to evaluate the performance of our proposed WaveTS in the long-term time series forecasting (LTSF) task. These datasets include Weather, Exchange, Traffic, Electricity (ECL), and ETT (four subsets). Additionally, 5 datasets are utilized for the short-term time series forecasting (STSF) task, including M4 and PEMS (four subsets). Detailed descriptions of these datasets can be found in Appendix C.1.

**Baselines** WaveTS is compared with several SOTA models from two categories (see details in Appendix C.2), including (1) FRL-based forecasters: FITS (Xu et al., 2024), DeRiTS (Fan et al., 2014), WPMixer (Murad et al., 2025) and TexFilter (Yi et al., 2024a). (2) Other forecasters: one hybrid model TimeMixer++ (Wang et al., 2025b), two representative Transformer-based models, including iTransformer (Liu et al., 2024c), PatchTST (Nie et al., 2023) and one MLP-based model DLinear (Zeng et al., 2023). We follow the official configurations when reproducing the results.

**Implementation Details** Our experiments are conducted using PyTorch on a single NVIDIA RTX A6000 GPU. We utilize the ADAM (Kingma & Ba, 2014) optimizer to minimize the backcasting while forecasting loss, as elaborated in Section 3.2.3. Furthermore, we adhere to the settings established in FITS (Xu et al., 2024) for WaveTS and the other nine baselines, considering the lookback length as a hyperparameter for grid search. This approach ensures a fair comparison, as previous literature has demonstrated that extended lookback windows optimize the performance of FRL-based forecasters (Yang et al., 2024), aligning with their training configurations. Detailed explanations of the evaluation metrics, MSE (Mean Squared Errors) and MAE (Mean Absolute Errors), are provided in Appendix C.3. The **db1** basis (Daubechies, 1988) function is uniformly selected across experiments due to its real-valued property and more analysis of the wavelet basis function in Appendix D.8.

## 4.2 Model Comparison on Time Series Forecasting

Table 1: Long-term forecasting MSE, results with MAE on more forecasting horizons are provided in Table 6.

| Dataset | WaveTS | FITS | DeRiTS | WPM | TexF | TMix+ | iTrans |
|---|---|---|---|---|---|---|---|
| ETTm1 | **0.356** | **0.361** | 0.715 | 0.365 | 0.391 | 0.377 | 0.407 |
| ETTm2 | **0.244** | **0.252** | 0.321 | 0.264 | 0.285 | 0.269 | 0.288 |
| ETTh1 | **0.410** | **0.412** | 0.682 | 0.418 | 0.441 | 0.419 | 0.454 |
| ETTh2 | **0.332** | **0.337** | 0.435 | 0.354 | 0.383 | 0.339 | 0.383 |
| ECL | **0.160** | 0.172 | 0.293 | 0.175 | 0.172 | **0.165** | 0.178 |
| Exchange | **0.361** | 0.458 | 0.427 | 0.426 | 0.388 | **0.357** | 0.362 |
| Traffic | **0.408** | 0.428 | 0.976 | 0.448 | 0.462 | **0.416** | **0.427** |
| Weather | 0.237 | **0.230** | 0.293 | 0.235 | 0.245 | **0.226** | 0.258 |
| Avg | **0.314** | 0.331 | 0.518 | 0.336 | 0.346 | **0.320** | 0.344 |

Table 2: Short-term forecasting MSE on four PEMS datasets, results with MAE are provided in Table 8.

| | Dataset | WaveTS | FITS | DeRiTS | WPM | TexF | iTrans | PatchT |
|---|---|---|---|---|---|---|---|---|
| Output =12 | PEMS03 | **0.075** | 1.041 | 0.239 | 0.083 | **0.076** | 0.083 | 0.097 |
| | PEMS04 | **0.091** | 1.116 | 0.206 | 0.098 | **0.090** | 0.101 | 0.105 |
| | PEMS07 | **0.071** | 1.080 | 0.170 | 0.080 | **0.073** | 0.077 | 0.095 |
| | PEMS08 | **0.091** | 1.375 | 0.276 | 0.096 | **0.087** | 0.095 | 0.168 |
| Output =24 | PEMS03 | **0.107** | 1.204 | 0.327 | 0.126 | **0.109** | 0.126 | 0.142 |
| | PEMS04 | **0.122** | 1.287 | 0.270 | 0.145 | **0.127** | 0.154 | 0.142 |
| | PEMS07 | **0.103** | 1.238 | 0.236 | 0.128 | **0.104** | 0.123 | 0.150 |
| | PEMS08 | **0.138** | 1.375 | 0.377 | 0.154 | **0.128** | 0.149 | 0.226 |
| | Avg | **0.100** | 1.215 | 0.263 | 0.114 | **0.099** | 0.114 | 0.141 |

Table 1 and Table 2 summarize the LTSF and STSF results, respectively, with the best in Red and the second in Blue[1]. Only MSE are reported here due to the space limit, and Avg represents the average of the results presented. Table 1 highlights the significant superiority of WaveTS in long-term forecasting especially on datasets characterized by challenging real-world scenarios with highly intricate temporal variations, such as ECL (Electricity) and Traffic, achieving overall MSE reduction of 5.1%. In short-term forecasting, WaveTS also demonstrates exceptional performance as summarized in Table 2. The PEMS datasets capture traffic time series data that exhibit complex fluctuations, irregular patterns, and sudden changes over time. Nonetheless, WaveTS achieves the best

[1]WPM (Murad et al., 2025), TexF (Yi et al., 2024a), TMix+ (Wang et al., 2025b), iTrans (Liu et al., 2024c), PatchT (Nie et al., 2023).

performance on these datasets, underscoring its effectiveness in leveraging non-stationary frequency features for time series forecasting. This paper also conducts experiments on the well-recognized M4 dataset for short-term forecasting task, the results are summarized in Table 9 in Appendix D.2.

## 4.3 MODEL ANALYSIS

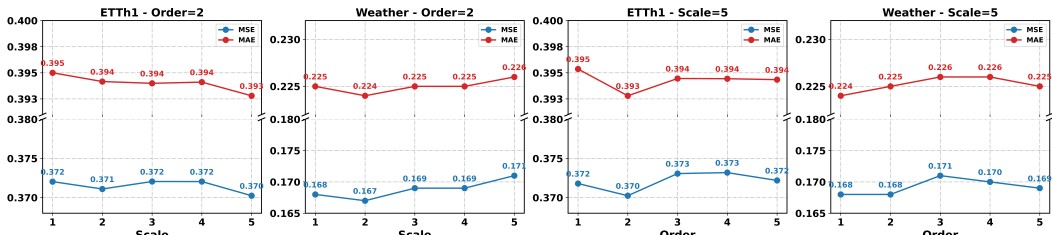

Figure 3: Model performance variations under different scales and orders in the WDT module. Results are collected from forecasting 96 experiments, each using its optimal input length.

**Hyper-parameter Analysis** The effectiveness of the WDT is intrinsically linked to two crucial hyperparameters, i.e. the **order** of derivative in WDT and the **scale** of the wavelet transform decomposition. The former parameter dictates the number of branches, influencing the whole architecture of the WaveTS while the latter influences the hierarchical multi-scale patterns learned within each branch. Consequently, we undertake experiments to assess hyper-parameter sensitivity using the ETTh1 and Weather datasets. Figure 3 delineates the influence that these two parameters exert on achieving optimal forecasting results. Generally, the performance of the WaveTS is not sensitive to the variations in order and scale, underscoring the robustness of the WaveTS. Additionally, a larger order or scale does not guarantee better performance; therefore, we adopt a case-by-case strategy in experiments for each forecasting horizon across different datasets. We provide numerical results on more datasets in Appendix D.5. In Appendix D.6, we also justify the principle for selecting hyperparameters with more experiments. In Appendix D.7, we quantitatively analyze the contribution of each WaveTS block to the final prediction and identify the **order** and **scale** configurations most suitable for different dataset types. Additionally, the selection of the **wavelet basis function** is also crucial in wavelet-based methods; therefore, we make a detailed analysis in Appendix D.8.

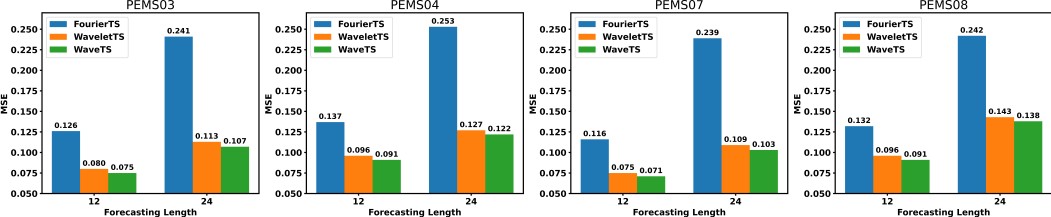

Figure 4: Ablation studies of the WDT on PEMS datasets. FourierTS and WaveletTS denote the substitution of the WDT with the Discrete Fourier Transform and the Discrete Wavelet Transform based on the WaveTS.

**Ablation Study of the WDT** As previously mentioned, our proposed WaveTS, is centered on the WDT. Consequently, we conduct an ablation study to validate the necessity of the WDT for effective time series modeling. Specifically, we replace the WDT with both the Wavelet Transform (WT) and the Fourier Transform (FT), along with their corresponding inverse transformations, also maintain the instance normalization and subsequent Frequency Refinement Units unchanged, obtaining WaveletTS and FourierTS for comparison. Detailed description can be found in Figure 10 in Appendix D.9. As demonstrated in Figure 4, the omission of the WDT leads to a significant decline in forecasting performance, thereby underscoring its essential role within the WaveTS framework. Moreover, the replacement of the WDT with the FT yields inferior performance when compared to the WT, indicating that the wavelet transform is more effective for time series analysis in FRL-based forecasters. We present additional experiments in Appendix D.9 with more results on several long-term forecasting benchmark datasets that reinforce the conclusions summarized in Figure 4.

**Visualizations of the Abrupt Changes Amplification** As illustrated in Figure 5, across all four datasets, the highlighted windows show that the magnitude of the wavelet derived coefficients surges precisely when the original series experiences a swift trend break and volatility jump. This

behavior is consistent with the analytical role of the derivative operation as a local high-pass filter: it suppresses slow-moving components while accentuating rapid derivatives. By contrast, the standard WT coefficients remain comparatively smooth, indicating limited sensitivity to such localized shocks. The repetition of this pattern on time series collected from various domain demonstrates that the WDT generalizes beyond a single domain, providing a robust mechanism for exposing change points that would otherwise be under-represented in frequency representations.

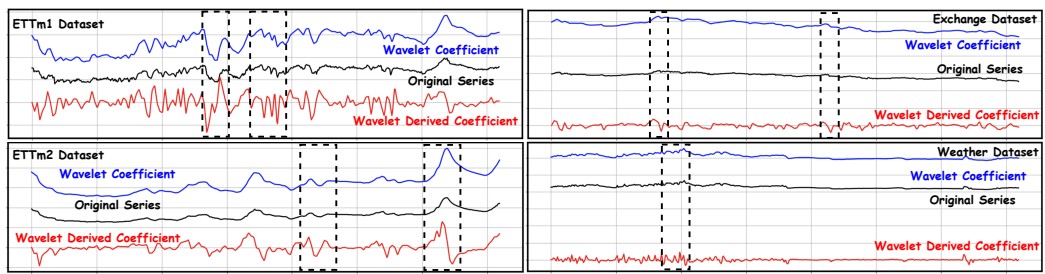

Figure 5: Additional visual evidence that the Wavelet Derived Coefficients (red) produced by the proposed WDT amplify abrupt regime shifts more clearly than the standard Wavelet Coefficients (blue).

**Visualization of Time-Frequency Scalograms**   Scalograms provide a robust framework for illustrating time-frequency characteristics within wavelet analysis, emphasizing the distribution of signal energy across various scales. In Figure 6, we present two examples at scale=5 from the ETTh1 dataset after 10 training epochs. Each row corresponds to a particular frequency band, with the lowest frequency component (LL) at the bottom and successively higher frequency details (LH1 to LH5) stacked above. Specifically, each WDT branch learns a distinct energy distribution, thus extracting different hierarchies of patterns from time series data and enhancing the model's representational capacity. In other words, same time step may exhibit differing energy levels across different orders of derivation. Collectively, these multi-scale branches capture both subtle changes and global patterns, contributing to improved time series modeling. More visualizations are provided in Appendix E.3.

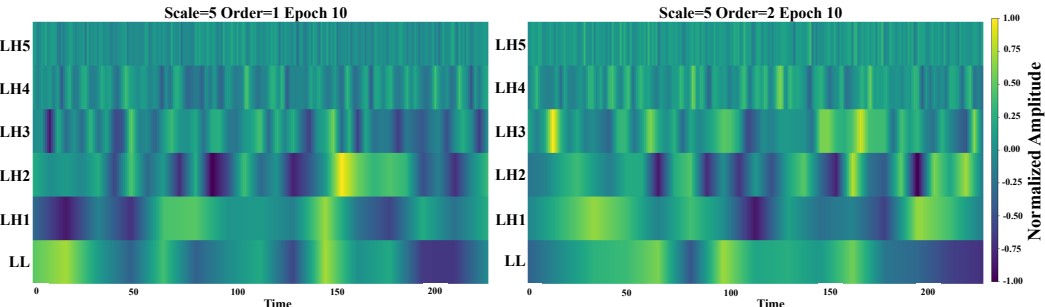

Figure 6: Visualization of scalograms in the WDT. The color intensity represents the normalized amplitude of the wavelet coefficients, indicating how different scales capture localized energy variations.

**Efficiency Analysis**   Table 3 displays the comparative results in terms of efficiency. "Train" and "Infer." refer to training time per epoch and inference time per batch, respectively. WaveTS demonstrates superior computational efficiency than other baseline methods while maintaining accurate forecasts. Specifically, WaveTS requires only 380MB of GPU memory, takes 2.61s per epoch for training, and achieves a remarkably low inference time of 0.015s per batch, all while obtaining an MSE of 0.303, surpassing both existing FRL-based and Transformer-based approaches in overall performance. WaveTS strikes an optimal balance between forecast accuracy and hardware usage: its efficiency gains

Table 3: Efficiency summary of WaveTS and other baselines on the ETTm1 dataset with a forecast horizon of 96. The lookback length is 360 and batch size is 64. Other configurations are set as recommended.

| Model | Memory | Train | Infer. | MSE |
|---|---|---|---|---|
| FEDformer (Zhou et al., 2022b) | 7666 MB | 52.40s | 0.152s | 0.325 |
| iTransformer (Liu et al., 2024c) | 474 MB | 5.23s | 0.053s | 0.314 |
| PatchTST (Nie et al., 2023) | 3360 MB | 25.11s | 0.076s | 0.325 |
| TimeMixer++ (Wang et al., 2025b) | 17506 MB | 30.21s | 0.104s | 0.310 |
| TexFilter (Yi et al., 2024a) | 450 MB | 3.34s | 0.021s | 0.321 |
| FreTS (Yi et al., 2023b) | 460 MB | 10.88s | 0.024s | 0.343 |
| DeRiTS (Fan et al., 2014) | 396 MB | 4.52s | 0.018s | 0.687 |
| FITS (Xu et al., 2024) | 394 MB | **2.46s** | 0.019s | 0.332 |
| WaveTS (Ours) | **380 MB** | 2.61s | **0.015s** | **0.303** |

come from the real-valued wavelet computations, which avoid the hardware-unfriendly nature of complex-valued operations common in prior FRL-based forecasters (Yi et al., 2023b; Xu et al., 2024; Zhou et al., 2022b; Cao et al., 2020; Yi et al., 2023a; Fan et al., 2014). This design allows WaveTS to deliver superior inference speed and low memory consumption, making it well-suited for practical deployment on edge devices and some resource-limited scenarios without sacrificing forecasting precision.

**Discussion**  Our proposed WaveTS expands the LTSF and the STSF toolbox and promotes the development of efficient, robust and scalable forecasting solutions through time-frequency transformation. We also provide discussion on the limitations of WaveTS in Appendix F.

## 5  CONCLUSION AND FUTURE WORK

This paper introduces WaveTS, which employs a multi-branch architecture to achieve superior results in deep time series forecasting. In contrast to previous forecasting models that utilize the Fourier transform for feature extraction, WaveTS targets modeling the derivative of time series rather than the raw data, effectively captures multi-scale time-frequency features, highlighting regime changes in time series through a theoretically sound Wavelet Derivative Transform, while preserving both information and energy. Empirical results on long-term forecasting task and short-term forecasting task demonstrate that WaveTS outperforms previous FRL-based and Transformer-based methods, concurrently maintaining low computational complexity. Future work will aim to optimize the WaveTS architecture for real-world deployment and broaden its applicability to downstream tasks, including classification and anomaly detection.

## REPRODUCIBILITY STATEMENT

We have taken several steps to ensure the reproducibility of our work. The model architecture is described in Section 3.2 of the main text. The theoretical proofs of our proposed algorithm are provided in Appendix B. The dataset statistics and descriptions are given in Appendix C.1. The implementation details, including training settings and hyperparameters, are provided in Appendix C.3. Furthermore, we include an anonymous link to the source code and scripts in the Abstract, which enables independent reproduction of our results.

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

# Appendix (Supplementary Material)

Multi-Order Wavelet Derivative Transform for Deep Time Series Forecasting

## TABLE OF CONTENTS

# A FREQUENCY DOMAIN REPRESENTATION LEARNING

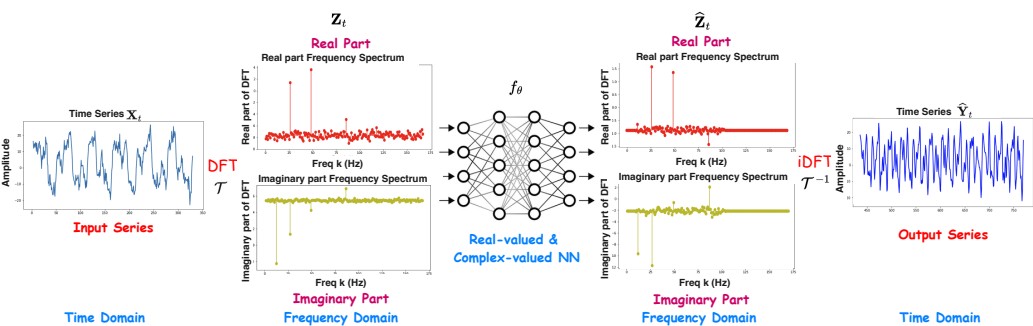

Figure 7: The paradigm of Frequency domain Representation Learning. Domain transformation is performed by the Discrete Fourier Transform (DFT) here.

We first clarify the definition of Frequency domain Representation Learning (FRL) for better understanding. Let $[X_1, X_2, \ldots, X_T] \in \mathbb{R}^{T \times C}$ denote a multivariate time series, where each $X_t \in \mathbb{R}^C$ contains the observations of $C$ variates at time $t$. At time $t$, the input window is $\mathbf{X}_t = [X_{t-L+1}, \ldots, X_t] \in \mathbb{R}^{L \times C}$ and the forecasting target is $\mathbf{Y}_t = [\hat{X}_{t+1}, \ldots, \hat{X}_{t+\tau}] \in \mathbb{R}^{\tau \times C}$. FRL models this map in the frequency domain: a bijective transform $\mathcal{T}$ produces coefficients $\mathbf{Z}_t = \mathcal{T}(\mathbf{X}_t)$; a learnable operator $f(\cdot \,; \theta)$ acts on the coefficients to yield $\widehat{\mathbf{Z}}_t = f(\mathbf{Z}_t \,; \theta)$; the inverse transform reconstructs the prediction $\widehat{\mathbf{Y}}_t = \mathcal{T}^{-1}(\widehat{\mathbf{Z}}_t)$ (see Figure 7). In the Fourier case, the coefficients comprise real and imaginary parts, or magnitude and phase; in the wavelet case, they comprise an $LL$ trend and multilevel $\mathbf{LH}$ details. Working on coefficients separates scales and oscillations, mitigates nonstationarity, enables band aware filtering and denoising, and remains lightweight because $f$ operates on compact structured representations. The effectiveness of FRL is that the frequency spectrum exposes seasonality and other global regularities, and provides a holistic view of temporal behavior while supporting multi scale and multi frequency analysis Yi et al. (2024b). Therefore, FRL has been widely integrated across deep backbones for time series forecasting, including Transformer based designs (Zhou et al., 2022b; Eldele et al., 2024), graph neural networks (Cao et al., 2020; Yi et al., 2023a), and MLP style architectures (Yi et al., 2023b; Fan et al., 2014; Xu et al., 2024; Fei et al., 2025). In this paper, WaveTS instantiates FRL with the WDT, which is real valued with an exact inverse: we transform $\mathbf{X}_t$ to WDT coefficients, apply a compact branch wise mapping across derivative order and multilevel scale, then reconstruct $\widehat{\mathbf{Y}}_t$ by $\mathcal{T}^{-1}$.

# B THEORETICAL PROOFS

## B.1 WAVELET DERIVATIVE TRANSFORM

**Lemma B.1.** *Given a time series $X(t)$, the wavelet transform of the $n$ order derivative of $X(t)$, namely Wavelet Derivative Transform (WDT), is related to the wavelet transform of $X(t)$ using the $n$-th derivative of the mother wavelet $\psi^{(n)}(t)$ as follows:*

$$\text{WDT}_k^{(n)} = (-1)^n 2^{nk} \tilde{W}_k^{(n)}, \tag{8}$$

*where $\tilde{W}_k^{(n)}$ is the wavelet transform of $X(t)$ using the $n$-th derivative of the mother wavelet $\psi^{(n)}(t)$:*

$$\tilde{W}_k^{(n)} = \sum_{t=0}^{T-1} X(t) \, \psi_k^{(n)}(t). \tag{9}$$

*Here $\psi_k^{(n)}(t)$ is the $n$-th derivative of the discrete wavelet basis function, defined as:*

$$\psi_k^{(n)}(t) = 2^{k/2} \psi^{(n)}(2^k t - j), \tag{10}$$

*with $k$ indexing the scale and $j$ the translation.*

*Proof.* Starting with the definition of the wavelet transform of the $n$-th derivative of $X(t)$:

$$W_{X^{(n)}}(k, j) = \int_{-\infty}^{+\infty} X^{(n)}(t) \psi_{k,j}(t) dt. \tag{11}$$

Using integration by parts $n$ times, and assuming that boundary terms vanish (i.e., the boundary term $\sum_{m=0}^{n-1}(-1)^m\left[X^{(n-1-m)}(t)\,\psi_{k,j}^{(m)}(t)\right]_{-\infty}^{+\infty}$ sum to zero. In practice, this assumption is typically satisfied because many mother wavelets, such as Daubechies wavelets, are either compactly supported or decay rapidly (Daubechies, 1988; 1992), making $\psi_{k,j}^{(m)}(t)$ negligible at the boundaries. We then transfer the derivative from $X^{(n)}(t)$ to the wavelet function:

$$W_{X^{(n)}}(k,j) = (-1)^n \int_{-\infty}^{+\infty} X(t)\frac{d^n}{dt^n}\psi_{k,j}(t)dt. \tag{12}$$

Calculating the $n$-th derivative of the discrete wavelet basis function:

$$\frac{d^n}{dt^n}\psi_{k,j}(t) = 2^{k/2}\frac{d^n}{dt^n}\psi(2^k t - j) = 2^{k/2}\left((2^k)^n\psi^{(n)}(2^k t - j)\right) = 2^{k/2+nk}\psi^{(n)}(2^k t - j). \tag{13}$$

Therefore,

$$\begin{aligned} W_{X^{(n)}}(k,j) &= (-1)^n \int_{-\infty}^{+\infty} X(t)\frac{d^n}{dt^n}\psi_{k,j}(t)dt \\ &= (-1)^n 2^{k/2+nk}\int_{-\infty}^{+\infty} X(t)\psi^{(n)}(2^k t - j)dt. \end{aligned} \tag{14}$$

Recall that the wavelet transform of $X(t)$ using the $n$-th derivative of the mother wavelet is:

$$\tilde{W}_{k,j}^{(n)} = 2^{k/2}\int_{-\infty}^{+\infty} X(t)\psi^{(n)}(2^k t - j)dt. \tag{15}$$

Thus,

$$\int_{-\infty}^{+\infty} X(t)\psi^{(n)}(2^k t - j)dt = \frac{1}{2^{k/2}}\tilde{W}_{k,j}^{(n)}. \tag{16}$$

Substituting back into the Eq. 14:

$$W_{X^{(n)}}(k,j) = (-1)^n 2^{k/2+nk}\left(\frac{1}{2^{k/2}}\tilde{W}_{k,j}^{(n)}\right) = (-1)^n 2^{nk}\tilde{W}_{k,j}^{(n)}. \tag{17}$$

$\square$

This result establishes the theoretical relationship between the wavelet transform of the $n$-th derivative of a time series and the wavelet transform conducted with the $n$-th derivative of the mother wavelet. **To the best of our knowledge, this is the first theoretical analysis devoted to wavelet derivative learning**. In practice, however, one need not explicitly differentiate the mother wavelet function. Although the derivative formally relies on $\psi^{(n)}$, standard discrete wavelet filters (such as Daubechies (Daubechies, 1992)) already capture difference and derivative features at fine scales. Hence, one can simply apply the usual WT to the original signal and then scale the resulting coefficients (by factors depending on the scale and derivative order) to emulate higher-order derivatives. This avoids numerical differentiation and the complexity of directly constructing or discretizing $\psi^{(n)}$. Wavelets with sufficient vanishing moments inherently act as finite-difference operators, thereby providing a natural approximation to derivative information. The linearity and multi-scale nature of the wavelet transform make it especially attractive for derivative-related tasks in signal processing (Mallat, 1989; Daubechies, 1992).

**Theorem B.2** (Energy Conservation of WDT). *Let $X(t) \in L^2(\mathbb{R})$ and $\psi(t)$ be an admissible mother wavelet (essentially has a well-defined wavelet transform, zero mean, and sufficient decay and vanishing moments to handle the boundary and derivative argument). For the $n$-th derivative of $X(t)$, denoted $X^{(n)}(t)$, its WT coefficients $W_{X^{(n)}}(k,j)$ (using $\psi_{k,j}$) satisfy:*

$$\sum_{k,j}\left|W_{X^{(n)}}(k,j)\right|^2 = \|X^{(n)}\|_2^2 \quad \text{(up to a constant factor dependent on the wavelet family)}.$$

*Thus, the total energy of $X^{(n)}(t)$ is preserved in its wavelet coefficient domain.*

*Proof.* From the standard WT Parseval relation (Parseval, 1806), we know that (up to a wavelet-dependent constant):

$$\sum_{k,j} |W_X(k,j)|^2 \;=\; \| X \|_2^2.$$

By integrating by parts $n$ times in the definition of $W_{X^{(n)}}(k,j)$ and assuming boundary terms vanish (due to $\psi$'s compact support and fast decay), one obtains the key identity:

$$W_{X^{(n)}}(k,j) \;=\; (-1)^n \, 2^{n\,k} \, \tilde{W}_{k,j}^{(n)},$$

where $\tilde{W}_{k,j}^{(n)}$ is the transform of $X$ using the $n$-th derivative of the mother wavelet $\psi^{(n)}(t)$. Since $\psi^{(n)}$ typically remains admissible under suitable regularity assumptions, its associated wavelet transform also obeys a Parseval-type property. Consequently,

$$\sum_{k,j} |W_{X^{(n)}}(k,j)|^2 \;=\; \sum_{k,j} |(-1)^n \, 2^{n\,k} \, \tilde{W}_{k,j}^{(n)}|^2 \;=\; \sum_{k,j} 2^{2\,n\,k} \, |\tilde{W}_{k,j}^{(n)}|^2.$$

This sum is proportional to $\| X \|_2^2$, but $(\mathrm{i}\omega)^n$ in the frequency domain corresponds to $X^{(n)}(t)$ in the time domain, implying:

$$\| X^{(n)} \|_2^2 \;\sim\; \sum_{k,j} |W_{X^{(n)}}(k,j)|^2.$$

Hence the Wavelet Derivative Transform preserves the $L^2$ norm of $X^{(n)}(t)$, completing the proof. $\quad\square$

### B.2 LTI State Space Models are FRL-based Forecasters

State space models (SSMs) are highly effective for time series modeling due to their inherent simplicity and capacity to capture complex autoregressive dependencies. By integrating observed data with latent state variables, SSMs can accurately delineate dynamic patterns and the underlying processes present within the data (Behrouz et al., 2024; Hu et al., 2024b;a). This paper posits that SSMs serve as FRL-based forecasters in certain contexts; therefore, the **forecasting while backcasting** training approach consistently exhibits superior performance when applied to SSMs, corroborating the findings presented in Table 10. Here, we present a two-stage proof demonstrating a direct correspondence between Linear Time-Invariant (LTI) SSMs and frequency-domain linear mappings utilized by FRL-based forecasters.

**LTI SSM is a Global Convolution in the time domain** Consider a 1D LTI SSM of the form:

$$\mathbf{x}_{k+1} = A\,\mathbf{x}_k + B\,u_k \,, \; y_k = C\,\mathbf{x}_k + D\,u_k, \tag{18}$$

where $\mathbf{x}_k \in \mathbb{R}^d$ is the hidden state at time $k$, $u_k \in \mathbb{R}$ is the input, and $y_k \in \mathbb{R}$ is the output. Matrices $A, B, C$ (and scalar $D$) remain constant over time. By iterating the first equation from some initial state $\mathbf{x}_0$, one obtains:

$$\mathbf{x}_k = A^k\mathbf{x}_0 \;+\; \sum_{n=0}^{k-1} A^{k-1-n}\,B\,u_n, \tag{19}$$

and substituting into the second equation yields:

$$y_k = C\Big( A^k\mathbf{x}_0 + \sum_{n=0}^{k-1} A^{k-1-n}B\,u_n \Big) \;+\; D\,u_k. \tag{20}$$

Ignoring the $C\,A^k\mathbf{x}_0$ term (for large $k$ or with $\|A\| < 1$ ensuring stability) leads to:

$$y_k = \sum_{\ell=0}^{\infty} \big(\overline{\mathbf{K}}\big)_\ell \, u_{k-1-\ell} \;+\; D\,u_k, \quad \overline{\mathbf{K}} = \big( C\,B, \; C\,A\,B, \; C\,A^2\,B, \; \ldots \big). \tag{21}$$

Thus, an LTI SSM is a **global convolution** of the input sequence $\{u_n\}$ with kernel $\overline{\mathbf{K}}$ (plus a direct feedthrough term $D\,u_k$).

**Global Convolution is Frequency-domain Linear Mapping**   Since a global convolution is the central time-domain operation of an LTI SSM, we next show that this same operation translates into a simple, elementwise linear mapping in the frequency domain. Concretely, let $\mathbf{H}(v)$, $W(v)$, and $B(v)$ be discrete-time sequences indexed by $v \in \mathbb{Z}$ (the time index). Think of $\mathbf{H}(v)$ as the convolution kernel (e.g., the impulse response), $W(v)$ as the input signal, and $B(v)$ as a bias or aforementioned "direct feedthrough" term. We denote by $\mathcal{H}(f)$, $\mathcal{W}(f)$, and $\mathcal{B}(f)$ their corresponding frequency-domain representations (e.g., via the Discrete Fourier Transform). In the time domain, the output of interest is:

$$\mathbf{H}(v) * W(v) \ + \ B(v). \tag{22}$$

By the Convolution Theorem (Oppenheim et al., 1999), we have:

$$\mathcal{F}\Big(\mathbf{H}(v) * W(v)\Big) = \mathcal{H}(f)\,\mathcal{W}(f) \text{ and } \mathcal{F}\Big(B(v)\Big) = \mathcal{B}(f). \tag{23}$$

Therefore,

$$\mathbf{H}(v) * W(v) + B(v) \ \longleftrightarrow \ \mathcal{H}(f)\,\mathcal{W}(f) + \mathcal{B}(f). \tag{24}$$

Combining these observations, we see that LTI SSMs—being global convolutions in the time domain—correspond precisely to linear operations in the frequency domain. In other words, under our definition of FRL-based forecaster as a frequency-domain linear mapping, LTI SSMs are indeed FRL-based forecasters. **This novel perspective provides additional insight into the efficacy of joint forecasting while backcasting training for SSM-based time series forecasters, as it aligns naturally with the frequency-domain interpretation of linear operators.**

## C   EXPERIMENTAL DETAILS

### C.1   DATASETS

We evaluate WaveTS using 10 real-world datasets, with detailed information provided in Table 4 for long-term series forecasting and Table 5 for short-term series forecasting. Specifically, the "Dimension" column indicates the number of variables in each dataset. "Forecasting Length" specifies the number of future time points to forecast, with four different forecasting horizons included for each dataset. "Dataset Size" represents the total number of time points in the training, validation, and test splits. "Information (Frequency)" denotes the domain and the sampling interval of the time series. Specifically, **ETT** comprises seven electricity transformer variables collected from July 2016 to July 2018. We utilize four subsets: ETTh1 and ETTh2 are sampled hourly, while ETTm1 and ETTm2 have 15-minute intervals. **Exchange** includes daily exchange rate data from eight countries spanning 1990 to 2016. **Weather** consists of 21 meteorological factors recorded every 10 minutes at the Weather Station of the Max Planck Bio-geochemistry Institute in 2020. **Electricity** captures hourly electricity consumption data from 321 clients. **Traffic** contains hourly road occupancy rates measured by 862 sensors on San Francisco Bay area freeways from January 2015 to December 2016. Additionally, we incorporate the **PEMS03**, **PEMS04**, **PEMS07**, and **PEMS08** datasets, which are traffic datasets from the California Transportation Agencies, capturing various aspects of traffic flow and congestion. The **M4** dataset comprises 100,000 time series from diverse domains, including finance, economics, and demographics, providing a comprehensive benchmark for evaluating forecasting methods across multiple scales and frequencies. Datasets are provided from (Wu et al., 2023; Zhou et al., 2021). All datasets undergo the same preprocessing steps and are split into training, validation, and test sets following the chronological order to prevent data leakage, as outlined in TimesNet Wu et al. (2023).

Table 4: Detailed dataset descriptions in long-term forecasting.

| Dataset | Dimension | Forecasting Length | Dataset Size | Information (Frequency) |
|---|---|---|---|---|
| ETTm1 | 7 | {96, 192, 336, 720} | (34369, 11425, 11425) | Electricity (15 min) |
| ETTh1 | 7 | {96, 192, 336, 720} | (8445, 2785, 2785) | Electricity (Hourly) |
| ETTm2 | 7 | {96, 192, 336, 720} | (34369, 11425, 11425) | Electricity (15 min) |
| ETTh2 | 7 | {96, 192, 336, 720} | (8545, 2881, 2881) | Electricity (Hourly) |
| Exchange | 8 | {96, 192, 336, 720} | (5120, 665, 1422) | Exchange rate (Daily) |
| Weather | 21 | {96, 192, 336, 720} | (36696, 5175, 10440) | Weather (10 min) |
| Electricity | 321 | {96, 192, 336, 720} | (18221, 2537, 5165) | Electricity (Hourly) |
| Traffic | 862 | {96, 192, 336, 720} | (12089, 1661, 3413) | Transportation (Hourly) |

Table 5: Detailed dataset descriptions in short-term forecasting.

| Dataset | Dimension | Forecasting Length | Dataset Size | Information (Frequency) |
|---|---|---|---|---|
| PEMS03 | 358 | {12, 24} | (15617, 5135, 5135) | Transportation (5min) |
| PEMS04 | 307 | {12, 24} | (10172, 3375, 3375) | Transportation (5min) |
| PEMS07 | 883 | {12, 24} | (16911, 5622, 5622) | Transportation (5min) |
| PEMS08 | 170 | {12, 24} | (10690, 3548, 3548) | Transportation (5min) |
| M4-Yearly | 1 | 6 | (23000, 0, 23000) | Various Domain Fusion |
| M4-Quarterly | 1 | 8 | (24000, 0, 24000) | Various Domain Fusion |
| M4-Monthly | 1 | 18 | (48000, 0, 48000) | Various Domain Fusion |
| M4-Weekly | 1 | 13 | (359, 0, 359) | Various Domain Fusion |
| M4-Daily | 1 | 14 | (4227, 0, 4227) | Various Domain Fusion |
| M4-Hourly | 1 | 58 | (414, 0, 414) | Various Domain Fusion |

## C.2 BASELINES

**FITS Xu et al. (2024):** FITS is a lightweight model for time series analysis that operates by interpolating time series in the complex frequency domain. This approach enables accurate forecasting and anomaly detection with only about 10k parameters, making it suitable for edge devices. We download the official code from: https://github.com/VEWOXIC/FITS.

**DeRiTS Fan et al. (2014):** DeRiTS addresses distribution shifts in non-stationary time series by utilizing the entire frequency spectrum through Frequency Derivative Transformation (FDT) and an Order-adaptive Fourier Convolution Network, enhancing forecasting performance and shift alleviation. Due to the absence of an open-source implementation of DeRiTS, we undertake the task of reproducing the code independently to replicate all experiments.

**TexFilter Yi et al. (2024a):** TexFilter enhances Transformer-based forecasting by incorporating learnable frequency filters that selectively pass or attenuate signal components. This method improves robustness to high-frequency noise and leverages the full frequency spectrum for more accurate predictions. We download the official code from: https://github.com/aikunyi/FilterNet.

**WPMixer Murad et al. (2025):** In WPMixer, multi-resolution wavelet transform extracts joint time–frequency features; the resulting sub-series are patched and embedded to extend history and preserve local patterns. An MLP-Mixer then globally mixes the patches, enabling the model to exploit both local and global information. However, a careful review of its official code revealed a fatal **drop last** bug (see line 23 in data_factory.py in https://github.com/Secure-and-Intelligent-Systems-Lab/WPMixer). Specifically, they excluded a significant portion of test data—especially with large batch sizes (128 and 256 in their experiments), resulting in unfair comparisons. Therefore, we fix the bug and rerun their experiments following the reported configurations.

**TimeMixer Wang et al. (2024a):** TimeMixer regards a time series as a set of disentangled fine-scale (seasonal) and coarse-scale (trend) signals, then fuses them bidirectionally: Past-Decomposable-Mixing (PDM) mixes seasonal and trend components from fine to coarse and coarse to fine, while Future-Multipredictor-Mixing (FMM) ensembles multiple predictors across those scales. We download the code from: https://github.com/kwuking/TimeMixer for short-term forecasting.

**TimeMixer++ Wang et al. (2025b):** TimeMixer++ is a versatile model that excels at various time series tasks by effectively capturing and extracting complex patterns across multiple time scales and frequency resolutions. Through techniques like multi-resolution time imaging and various mixing strategies, TimeMixer++ achieves state-of-the-art performance, outperforming both general-purpose and task-specific models in forecasting, classification, and other time series analyses. Although an GitHub repository implementing TimeMixer++ is available at https://anonymous.4open.science/r/TimeMixerPP , rerunning the code does not reproduce the results reported in the paper. Therefore, we adopt the results directly from the paper only for long-term forecasting task. For Table 3, however, we rerun the provided code to obtain the reported efficiency metrics.

**iTransformer Liu et al. (2024c):** iTransformer reconfigures standard Transformer models for time series forecasting by switching input dimensions and leveraging attention mechanisms across variate

tokens. This adaptation improves the model's capability to learn temporal patterns. We download the official code from: https://github.com/thuml/iTransformer.

**PatchTST Nie et al. (2023):** PatchTST modifies Transformer architectures with patch-based processing and a channel-independent framework, optimizing performance for long-term forecasting tasks. Its design is particularly effective for capturing intricate temporal structures. We download the official code from: https://github.com/PatchTST.

**TimesNet Wu et al. (2023):** TimesNet introduces a task-agnostic framework that reshapes one-dimensional time series into two-dimensional tensors, enabling it to address multi-periodic patterns. We download the official code from: https://github.com/thuml/Time-Series-Library.

**DLinear Zeng et al. (2023):** DLinear questions the suitability of Transformers for long-term time series forecasting and proposes a simple one-layer linear model (LTSF-Linear) as a baseline. Surprisingly, LTSF-Linear consistently outperforms sophisticated Transformer-based models across nine real-world datasets, highlighting the need to rethink research directions in time series analysis. We download the official code from: https://github.com/cure-lab/LTSF-Linear for long-term forecasting.

### C.3 IMPLEMENTATION DETAILS

All experiments are implemented in PyTorch (Paszke et al., 2019) and executed on a single NVIDIA RTX A6000 GPU (40 GB). Optimization is carried out with Adam (Kingma & Ba, 2014); the initial learning rate $\eta$ is sampled log-uniformly from $[10^{-5}, 10^{-2}]$, the loss defined in Section 3.2.3 is minimized for training with early stop mechanism, and the batch size is tuned over $\{16, 32, 64, 128\}$. Following FITS almost entirely, we use optuna (Akiba et al., 2019) to carefully tune three hyper-parameters—look-back window length from $[192, 1440]$, Wavelet decomposition scale $s \in \{1, 2, 3, 4, 5\}$, and Wavelet derivative order $k \in \{1, 2, 3, 4, 5\}$—choosing the best $L$ on the validation set to prevent information leakage, because longer contexts (e.g., $L = 336$) typically improve most models whereas iTransformer (Liu et al., 2024c) and FreTS (Yi et al., 2023b) are comparatively insensitive. After fixing the optimal hyper-parameters, we retrain the model and report its test performance, while all baselines are re-run with their officially recommended configurations to ensure a fair comparison.

**Long-Term Forecasting:** We employ a range of metrics specifically designed for the tasks at hand. In particular, for the ground truth $\mathbf{X}$ and predictions $\widehat{\mathbf{X}}$ of size $H \times C$, the MSE and MAE are defined as follows:

$$\text{MSE} = \frac{1}{HC} \sum_{i=1}^{H} \sum_{c=1}^{C} \left(\mathbf{X}_{i,c} - \widehat{\mathbf{X}}_{i,c}\right)^2, \ \text{MAE} = \frac{1}{HC} \sum_{i=1}^{H} \sum_{c=1}^{C} \left|\mathbf{X}_{i,c} - \widehat{\mathbf{X}}_{i,c}\right|.$$

**Short-Term Forecasting:** We utilize the symmetric mean absolute percentage error (SMAPE), mean absolute scaled error (MASE), and overall weighted average (OWA). Let $\mathbf{X}, \widehat{\mathbf{X}} \in \mathbb{R}^{H \times C}$ be the ground truth and predictions, and $\mathbf{X}_i$ the $i$-th future time point, then:

$$\text{SMAPE} = \frac{200}{H} \sum_{i=1}^{H} \frac{|\mathbf{X}_i - \widehat{\mathbf{X}}_i|}{|\mathbf{X}_i| + |\widehat{\mathbf{X}}_i|},$$

$$\text{MASE} = \frac{1}{H} \sum_{i=1}^{H} \frac{|\mathbf{X}_i - \widehat{\mathbf{X}}_i|}{\frac{1}{H-m} \sum_{j=m+1}^{H} |\mathbf{X}_j - \mathbf{X}_{j-m}|},$$

$$\text{OWA} = \frac{1}{2} \left( \frac{\text{SMAPE}}{\text{SMAPE}_{\text{Naïve2}}} + \frac{\text{MASE}}{\text{MASE}_{\text{Naïve2}}} \right).$$

where $m$ is the data periodicity. OWA is a specialized metric employed in the M4 competition (Makridakis et al., 2018), which facilitates the comparison of SMAPE and MASE against a Naïve2 baseline.

# D ADDITIONAL RESULTS

## D.1 LONG-TERM FORECASTING

Table 6 presents the long-term forecasting performance of our proposed WaveTS model compared to various contemporary methodologies across multiple datasets and forecasting horizons (96, 192, 336, 720). The reported results are averaged over 5 runs with different random seeds (see standard deviations in Table 7). **Two-tailed paired Student's t-tests ($\alpha = 0.05$) applied to the per-run metrics show that WaveTS's improvements compared with previous baselines are statistically significant for most datasets and forecasting horizons (all $p < 0.05$).** Specifically, WaveTS consistently outperforms existing FRL-based forecasters, such as FITS (Xu et al., 2024) and DeRiTS (Fan et al., 2014) achieving the best MSE and MAE in most datasets. Across all datasets, WaveTS maintains the lowest overall average MSE of **0.314** and MAE of **0.338**, demonstrating a substantial improvement of approximately 5.1% in MSE compared to the next best model, FITS (Xu et al., 2024). Additionally, WaveTS achieves top performance in the ETT datasets, further highlighting its robust ability to model complex temporal patterns.

Table 6: Full long-term forecasting results. Red: the best results, Blue: the second best.

| Models | | FRL-based Forecasters | | | | | | | | | Other Forecasters | | | | | | | |
|---|---|---|---|---|---|---|---|---|---|---|---|---|---|---|---|---|---|---|
| | | WaveTS (Ours) | | FITS (2024) | | DeRiTS (2014) | | WPMixer (2025) | | TexFilter (2024a) | | TimeMixer++ 2025b | | iTransformer (2024c) | | PatchTST (2023) | | DLinear (2023) |
| Metrics | | MSE MAE | | MSE MAE | | MSE MAE | | MSE MAE | | MSE MAE | | MSE MAE | | MSE MAE | | MSE MAE | | MSE MAE |
| ETTm1 | 96 | 0.301 0.344 | | 0.306 0.348 | | 0.691 0.541 | | 0.309 0.346 | | 0.321 0.361 | | 0.313 0.354 | | 0.334 0.368 | | 0.329 0.367 | | 0.346 0.374 |
| | 192 | 0.338 0.365 | | 0.340 0.369 | | 0.708 0.550 | | 0.350 0.369 | | 0.367 0.387 | | 0.356 0.380 | | 0.377 0.391 | | 0.367 0.385 | | 0.382 0.391 |
| | 336 | 0.367 0.384 | | 0.373 0.388 | | 0.719 0.558 | | 0.372 0.394 | | 0.401 0.409 | | 0.386 0.402 | | 0.426 0.420 | | 0.399 0.410 | | 0.415 0.415 |
| | 720 | 0.416 0.412 | | 0.424 0.419 | | 0.742 0.572 | | 0.430 0.422 | | 0.477 0.448 | | 0.452 0.437 | | 0.491 0.459 | | 0.454 0.439 | | 0.473 0.451 |
| | Avg | 0.356 0.376 | | 0.361 0.381 | | 0.715 0.555 | | 0.365 0.383 | | 0.391 0.401 | | 0.377 0.393 | | 0.407 0.410 | | 0.387 0.400 | | 0.404 0.408 |
| ETTm2 | 96 | 0.162 0.252 | | 0.165 0.256 | | 0.227 0.308 | | 0.170 0.254 | | 0.175 0.258 | | 0.170 0.245 | | 0.180 0.264 | | 0.175 0.259 | | 0.193 0.293 |
| | 192 | 0.215 0.292 | | 0.219 0.294 | | 0.284 0.338 | | 0.228 0.293 | | 0.240 0.301 | | 0.229 0.291 | | 0.250 0.309 | | 0.241 0.302 | | 0.284 0.361 |
| | 336 | 0.263 0.326 | | 0.271 0.328 | | 0.339 0.370 | | 0.290 0.330 | | 0.311 0.347 | | 0.303 0.343 | | 0.311 0.348 | | 0.305 0.343 | | 0.382 0.429 |
| | 720 | 0.335 0.373 | | 0.352 0.382 | | 0.434 0.419 | | 0.367 0.390 | | 0.414 0.405 | | 0.373 0.399 | | 0.412 0.407 | | 0.402 0.400 | | 0.558 0.525 |
| | Avg | 0.244 0.311 | | 0.252 0.315 | | 0.321 0.359 | | 0.264 0.317 | | 0.285 0.328 | | 0.269 0.320 | | 0.288 0.332 | | 0.281 0.326 | | 0.354 0.402 |
| ETTh1 | 96 | 0.367 0.391 | | 0.374 0.396 | | 0.625 0.531 | | 0.368 0.394 | | 0.382 0.402 | | 0.361 0.403 | | 0.386 0.405 | | 0.374 0.399 | | 0.397 0.412 |
| | 192 | 0.404 0.414 | | 0.407 0.416 | | 0.665 0.550 | | 0.419 0.419 | | 0.430 0.429 | | 0.416 0.441 | | 0.441 0.436 | | 0.417 0.422 | | 0.446 0.441 |
| | 336 | 0.427 0.432 | | 0.430 0.436 | | 0.710 0.574 | | 0.438 0.433 | | 0.472 0.451 | | 0.430 0.434 | | 0.487 0.458 | | 0.431 0.436 | | 0.489 0.467 |
| | 720 | 0.440 0.455 | | 0.435 0.458 | | 0.730 0.608 | | 0.446 0.460 | | 0.481 0.473 | | 0.467 0.451 | | 0.503 0.491 | | 0.445 0.445 | | 0.513 0.510 |
| | Avg | 0.410 0.423 | | 0.412 0.427 | | 0.682 0.566 | | 0.418 0.427 | | 0.441 0.439 | | 0.419 0.432 | | 0.454 0.448 | | 0.417 0.435 | | 0.461 0.457 |
| ETTh2 | 96 | 0.267 0.333 | | 0.273 0.339 | | 0.380 0.400 | | 0.281 0.336 | | 0.293 0.343 | | 0.276 0.328 | | 0.297 0.349 | | 0.302 0.348 | | 0.340 0.394 |
| | 192 | 0.332 0.375 | | 0.334 0.377 | | 0.442 0.435 | | 0.350 0.380 | | 0.374 0.396 | | 0.342 0.379 | | 0.380 0.400 | | 0.388 0.400 | | 0.482 0.479 |
| | 336 | 0.349 0.396 | | 0.356 0.398 | | 0.465 0.461 | | 0.374 0.405 | | 0.417 0.430 | | 0.346 0.398 | | 0.428 0.432 | | 0.426 0.433 | | 0.591 0.541 |
| | 720 | 0.380 0.428 | | 0.384 0.427 | | 0.452 0.459 | | 0.412 0.432 | | 0.449 0.460 | | 0.392 0.415 | | 0.427 0.445 | | 0.431 0.446 | | 0.839 0.661 |
| | Avg | 0.332 0.383 | | 0.337 0.385 | | 0.435 0.439 | | 0.354 0.388 | | 0.383 0.407 | | 0.339 0.380 | | 0.383 0.407 | | 0.387 0.407 | | 0.563 0.519 |
| Electricity | 96 | 0.131 0.227 | | 0.145 0.242 | | 0.275 0.362 | | 0.148 0.240 | | 0.147 0.245 | | 0.135 0.222 | | 0.148 0.240 | | 0.181 0.270 | | 0.210 0.302 |
| | 192 | 0.146 0.240 | | 0.157 0.252 | | 0.277 0.364 | | 0.161 0.250 | | 0.160 0.251 | | 0.147 0.235 | | 0.162 0.253 | | 0.188 0.274 | | 0.210 0.305 |
| | 336 | 0.162 0.256 | | 0.174 0.269 | | 0.291 0.376 | | 0.177 0.265 | | 0.173 0.267 | | 0.164 0.245 | | 0.178 0.269 | | 0.204 0.293 | | 0.223 0.319 |
| | 720 | 0.200 0.288 | | 0.213 0.301 | | 0.329 0.402 | | 0.215 0.302 | | 0.210 0.309 | | 0.212 0.310 | | 0.225 0.317 | | 0.246 0.324 | | 0.258 0.350 |
| | Avg | 0.160 0.253 | | 0.172 0.266 | | 0.293 0.376 | | 0.175 0.264 | | 0.172 0.268 | | 0.165 0.253 | | 0.178 0.270 | | 0.205 0.290 | | 0.225 0.319 |
| Exchange | 96 | 0.086 0.204 | | 0.109 0.235 | | 0.143 0.271 | | 0.102 0.220 | | 0.091 0.211 | | 0.085 0.214 | | 0.088 0.208 | | 0.089 0.205 | | 0.088 0.218 |
| | 192 | 0.177 0.300 | | 0.229 0.350 | | 0.240 0.355 | | 0.202 0.310 | | 0.186 0.305 | | 0.175 0.313 | | 0.178 0.298 | | 0.179 0.299 | | 0.176 0.315 |
| | 336 | 0.322 0.411 | | 0.400 0.463 | | 0.387 0.456 | | 0.360 0.433 | | 0.380 0.449 | | 0.316 0.420 | | 0.331 0.417 | | 0.366 0.435 | | 0.313 0.427 |
| | 720 | 0.860 0.693 | | 1.095 0.781 | | 0.940 0.938 | | 1.041 0.923 | | 0.896 0.712 | | 0.851 0.689 | | 0.852 0.698 | | 0.901 0.714 | | 0.839 0.695 |
| | Avg | 0.361 0.402 | | 0.458 0.457 | | 0.427 0.505 | | 0.426 0.471 | | 0.388 0.421 | | 0.362 0.391 | | 0.362 0.405 | | 0.383 0.413 | | 0.354 0.414 |
| Traffic | 96 | 0.382 0.266 | | 0.401 0.280 | | 0.961 0.542 | | 0.431 0.312 | | 0.430 0.294 | | 0.392 0.253 | | 0.395 0.268 | | 0.462 0.295 | | 0.650 0.396 |
| | 192 | 0.394 0.270 | | 0.415 0.286 | | 0.973 0.547 | | 0.411 0.310 | | 0.452 0.307 | | 0.402 0.258 | | 0.416 0.276 | | 0.466 0.296 | | 0.598 0.370 |
| | 336 | 0.409 0.278 | | 0.429 0.290 | | 0.959 0.556 | | 0.443 0.311 | | 0.470 0.316 | | 0.428 0.263 | | 0.430 0.282 | | 0.482 0.304 | | 0.605 0.373 |
| | 720 | 0.447 0.298 | | 0.468 0.308 | | 1.010 0.556 | | 0.505 0.329 | | 0.498 0.323 | | 0.441 0.282 | | 0.466 0.305 | | 0.514 0.322 | | 0.645 0.394 |
| | Avg | 0.408 0.278 | | 0.428 0.291 | | 0.976 0.545 | | 0.448 0.316 | | 0.462 0.310 | | 0.416 0.264 | | 0.427 0.283 | | 0.481 0.304 | | 0.625 0.383 |
| Weather | 96 | 0.167 0.223 | | 0.145 0.199 | | 0.216 0.270 | | 0.168 0.205 | | 0.162 0.207 | | 0.155 0.205 | | 0.174 0.214 | | 0.177 0.218 | | 0.195 0.252 |
| | 192 | 0.210 0.258 | | 0.190 0.243 | | 0.264 0.304 | | 0.209 0.270 | | 0.210 0.250 | | 0.201 0.245 | | 0.221 0.254 | | 0.225 0.259 | | 0.237 0.295 |
| | 336 | 0.256 0.294 | | 0.238 0.282 | | 0.312 0.335 | | 0.263 0.289 | | 0.265 0.290 | | 0.237 0.265 | | 0.278 0.296 | | 0.278 0.297 | | 0.282 0.331 |
| | 720 | 0.315 0.336 | | 0.310 0.332 | | 0.380 0.375 | | 0.339 0.339 | | 0.342 0.340 | | 0.312 0.334 | | 0.358 0.347 | | 0.354 0.348 | | 0.345 0.382 |
| | Avg | 0.237 0.278 | | 0.230 0.266 | | 0.293 0.321 | | 0.235 0.283 | | 0.245 0.272 | | 0.226 0.262 | | 0.258 0.278 | | 0.259 0.280 | | 0.265 0.315 |
| Overall Avg | | 0.314 0.338 | | 0.331 0.348 | | 0.518 0.458 | | 0.336 0.356 | | 0.346 0.356 | | 0.320 0.335 | | 0.344 0.354 | | 0.350 0.357 | | 0.406 0.402 |

Table 7: Standard error results of WaveTS in long-term forecasting with the input length of 336.

| Tasks | ETTm1 | | ETTm2 | | ETTh1 | | ETTh2 | | Weather | | Electricity | | Traffic | | Exchange | |
|---|---|---|---|---|---|---|---|---|---|---|---|---|---|---|---|---|
| Metric | MSE | MAE | MSE | MAE | MSE | MAE | MSE | MAE | MSE | MAE | MSE | MAE | MSE | MAE | MSE | MAE |
| 96 | 2e-3 | 1e-3 | 2e-3 | 2e-3 | 1e-3 | 1e-3 | 1e-3 | 1e-3 | 2e-3 | 2e-3 | 1e-3 | 1e-3 | 3e-3 | 2e-3 | 1e-3 | 1e-3 |
| 192 | 2e-3 | 1e-3 | 1e-3 | 1e-3 | 2e-3 | 2e-3 | 1e-3 | 1e-3 | 2e-3 | 2e-3 | 2e-3 | 1e-3 | 3e-3 | 2e-3 | 2e-3 | 1e-3 |
| 336 | 2e-3 | 1e-3 | 2e-3 | 2e-3 | 1e-3 | 1e-3 | 1e-3 | 1e-3 | 2e-3 | 2e-3 | 3e-3 | 3e-3 | 3e-3 | 2e-3 | 3e-3 | 2e-3 |
| 720 | 3e-3 | 2e-3 | 4e-3 | 3e-3 | 3e-3 | 2e-3 | 5e-3 | 4e-3 | 3e-3 | 3e-3 | 4e-3 | 4e-3 | 5e-3 | 4e-3 | 3e-3 | 3e-3 |
| Avg | 2e-3 | 1e-3 | 2e-3 | 2e-3 | 2e-3 | 1e-3 | 4e-3 | 2e-3 | 2e-3 | 2e-3 | 2e-3 | 2e-3 | 3e-3 | 2e-3 | 2e-3 | 2e-3 |

Furthermore, it is worth emphasizing that our 5.1% reduction in MSE represents a significant leap forward. In many recently proposed forecasting methods (Yi et al., 2024a; Wang et al., 2024a; Hu

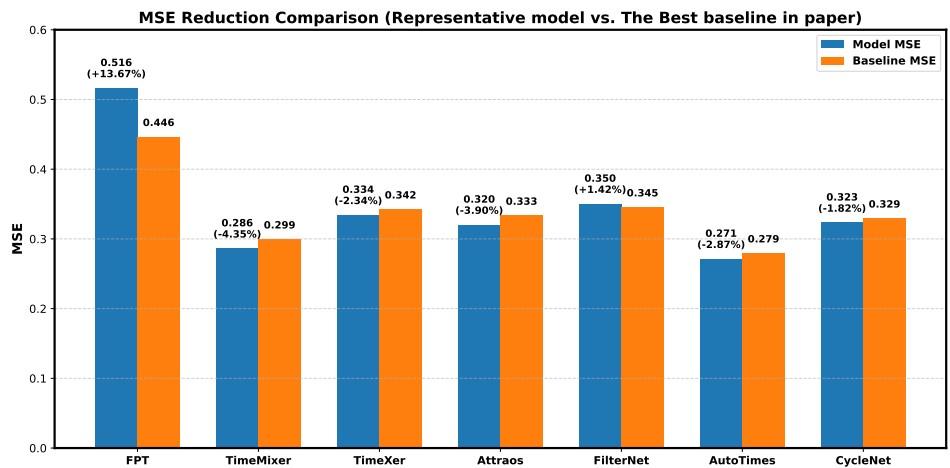

Figure 8: Comparison of MSE reduction of several recent SOTA deep time series forecasting models.

et al., 2024a; Wang et al., 2024d; Liu et al., 2024d; Lin et al., 2024), improvements over baselines tend to remain below the 5%, as illustrated in Figure 8, indicating relatively marginal performance gains. Notably, the values shown in Figure 8 reflect each model's average MSE improvement (or lack thereof) over its strongest baseline across a variety of datasets and forecasting horizons. In fact, some methods even fail to outperform the baseline when averaged over all experimental settings. **Surpassing this margin underscores that WaveTS's improvement is not marginal, highlighting its capacity in long-term forecasting.**

## D.2 SHORT-TERM FORECASTING

Table 8: Short-term forecasting results on PEMS datasets with forecasting horizon $\tau \in \{12, 24\}$.

| Models | | WaveTS | | FITS | | DeRiTS | | WPMixer | | TexFilter | | iTransformer | | PatchTST | |
|---|---|---|---|---|---|---|---|---|---|---|---|---|---|---|---|---|
| Metrics | | MSE | MAE | MSE | MAE | MSE | MAE | MSE | MAE | MSE | MAE | MSE | MAE | MSE | MAE |
| $O = 12$ | PEMS03 | **0.075** | **0.178** | 1.041 | 0.844 | 0.239 | 0.370 | 0.083 | 0.192 | **0.076** | **0.186** | 0.083 | 0.194 | 0.097 | 0.214 |
| | PEMS04 | **0.091** | **0.196** | 1.116 | 0.886 | 0.206 | 0.340 | 0.098 | 0.210 | **0.090** | 0.201 | 0.101 | 0.211 | 0.105 | 0.224 |
| | PEMS07 | **0.071** | **0.174** | 1.080 | 0.849 | 0.170 | 0.359 | 0.080 | 0.188 | **0.073** | **0.177** | 0.077 | 0.184 | 0.095 | 0.207 |
| | PEMS08 | **0.091** | **0.193** | 1.375 | 0.976 | 0.276 | 0.395 | 0.096 | 0.206 | **0.087** | **0.194** | 0.095 | 0.203 | 0.168 | 0.232 |
| $O = 24$ | PEMS03 | **0.107** | **0.208** | 1.204 | 0.915 | 0.327 | 0.432 | 0.126 | 0.240 | **0.109** | 0.225 | 0.126 | 0.241 | 0.142 | 0.259 |
| | PEMS04 | **0.122** | **0.228** | 1.287 | 0.956 | 0.270 | 0.387 | 0.145 | 0.258 | **0.127** | 0.242 | 0.154 | 0.266 | 0.142 | 0.259 |
| | PEMS07 | **0.103** | **0.209** | 1.238 | 0.916 | 0.236 | 0.362 | 0.128 | 0.241 | **0.104** | 0.216 | 0.123 | 0.236 | 0.150 | 0.262 |
| | PEMS08 | **0.138** | **0.231** | 1.375 | 0.976 | 0.377 | 0.462 | 0.154 | 0.264 | **0.128** | **0.237** | 0.149 | 0.258 | 0.226 | 0.284 |

Table 9: Full results for the short-term forecasting task in the M4 dataset.

| Models | | **WaveTS** | FITS | DeRiTS | WPMixer | TexFilter | iTransformer | PatchTST | FEDformer | TimesNet | TimeMixer |
|---|---|---|---|---|---|---|---|---|---|---|---|
| Yearly | SMAPE | **13.766** | 14.090 | 15.695 | 14.001 | 14.240 | 15.419 | 13.778 | 13.838 | 13.787 | **13.365** |
| | MASE | **2.919** | 3.366 | 3.346 | 3.040 | 3.109 | 3.218 | 2.985 | 3.048 | 3.134 | **2.926** |
| | OWA | **0.783** | 0.808 | 0.901 | 0.800 | 0.827 | 0.845 | 0.799 | 0.803 | 0.812 | **0.793** |
| Quarterly | SMAPE | **10.680** | 11.456 | 13.520 | 10.698 | 11.364 | 10.699 | 10.999 | 10.822 | 10.704 | **10.001** |
| | MASE | **1.288** | 1.302 | 1.645 | 1.212 | 1.328 | 1.289 | 1.293 | 1.382 | 1.390 | **1.168** |
| | OWA | 0.955 | 1.020 | 1.214 | 0.980 | 1.000 | 0.958 | **0.803** | 0.958 | 0.990 | **0.830** |
| Monthly | SMAPE | **13.379** | 14.302 | 15.985 | 13.623 | 14.014 | 16.650 | **12.641** | 14.262 | 13.670 | 13.544 |
| | MASE | **1.016** | 1.230 | 1.286 | 1.302 | 1.052 | 1.394 | 1.030 | 1.102 | 1.133 | **1.020** |
| | OWA | **0.942** | 1.011 | 1.159 | 0.950 | 0.981 | 1.232 | **0.946** | 1.012 | 0.978 | 0.997 |
| Others | SMAPE | **4.510** | 6.011 | 6.908 | 5.901 | 15.880 | 5.543 | 5.826 | 5.955 | 5.991 | **4.640** |
| | MASE | **3.114** | 4.007 | 4.507 | 4.001 | 11.434 | 3.998 | 4.610 | 3.268 | 3.613 | **3.210** |
| | OWA | **1.041** | 1.231 | 1.437 | 1.249 | 3.474 | 1.224 | 1.044 | 1.056 | 1.045 | **0.990** |
| Weighted Average | SMAPE | **12.442** | 13.165 | 14.875 | 13.423 | 13.525 | 14.153 | 14.821 | 12.850 | 12.829 | **11.729** |
| | MASE | **1.560** | 1.900 | 2.007 | 1.690 | 2.111 | 1.916 | 1.690 | 1.703 | 1.685 | **1.590** |
| | OWA | **0.840** | 0.901 | 1.073 | 0.904 | 1.023 | 1.051 | 0.901 | 0.918 | 0.900 | **0.850** |

Table 8 summarizes the short-term forecasting on four PEMS datasets with forecasting horizons range from 12 to 24. WaveTS achieves superior performance on these datasets, underscoring its effectiveness in leveraging non-stationary frequency features for enhanced time series forecasting. Table 9

showcases the short-term forecasting performance of our proposed WaveTS model in comparison with several state-of-the-art models on the M4 dataset. The evaluation spans multiple forecasting horizons, including Yearly, Quarterly, Monthly, and Others, utilizing metrics such as SMAPE, MASE, and OWA. In the **Yearly** category, WaveTS achieves the best MASE (**2.919**) and OWA (**0.783**) scores, indicating superior accuracy and reliability. These results collectively demonstrate that WaveTS not only maintains competitive performance across various forecasting horizons but also excels in multiple key metrics, validating its strong modeling capabilities for different time series.

## D.3 THE EFFECTIVENESS OF BACKCASTING WHILE FORECASTING

Table 10 compares each model trained only on the forecast window (F) with the same model trained to reconstruct the past and forecast the future (BF). MS is MambaSimple (Gu & Dao, 2024) for short). For every frequency-representation learner: WaveTS, FITS (Xu et al., 2024) and DeRiTS (Fan et al., 2014), adding the backcast term consistently lowers both MSE and MAE at all four horizons on every dataset. The same trend holds for the linear state-space model MambaSimple (Gu & Dao, 2024), whose frequency-domain stem is a diagonal (element-wise) linear filter; details and the proof that such SSMs are FRL-equivalent are given in Appendix B.2. The gain comes from an architectural match: FRL modules apply invertible, element-wise linear transforms in the frequency domain, so the transformed spectrum still contains a recoverable copy of the original coefficients. Training with BF forces the network to exploit that embedded history, providing a richer supervisory signal, stabilizing optimization, and ultimately producing more accurate forecasts.

Table 10: The effectiveness of backcasting and forecasting for FRL-based forecasters, (F) means only forecasting, (BF) means both backcasting and forecasting.

| Methods Metrics | | WaveTS(F) MSE | MAE | WaveTS(BF) MSE | MAE | FITS(F) MSE | MAE | FITS(BF) MSE | MAE | DeRiTS(F) MSE | MAE | DeRiTS(BF) MSE | MAE | MS(F) MSE | MAE | MS(BF) MSE | MAE |
|---|---|---|---|---|---|---|---|---|---|---|---|---|---|---|---|---|---|
| ETTh1 | 96 | 0.372 | 0.394 | **0.370** | **0.393** | 0.706 | 0.568 | **0.368** | **0.392** | 0.625 | 0.531 | **0.377** | **0.399** | 0.463 | 0.445 | **0.411** | **0.424** |
| | 192 | 0.405 | 0.415 | **0.404** | **0.413** | 0.713 | 0.575 | **0.405** | **0.412** | 0.665 | 0.550 | **0.407** | **0.417** | 0.504 | 0.494 | **0.467** | **0.480** |
| | 336 | 0.431 | 0.429 | **0.429** | **0.428** | 0.705 | 0.579 | **0.425** | **0.436** | 0.710 | 0.574 | **0.435** | **0.435** | 0.563 | 0.517 | **0.480** | **0.470** |
| | 720 | 0.441 | 0.457 | **0.436** | **0.454** | 0.701 | 0.596 | **0.431** | **0.451** | 0.730 | 0.608 | **0.467** | **0.477** | 0.538 | 0.508 | **0.533** | **0.505** |
| ETTh2 | 96 | 0.275 | 0.337 | **0.270** | **0.335** | 0.383 | 0.418 | **0.276** | **0.338** | 0.380 | 0.400 | **0.284** | **0.346** | 0.366 | 0.393 | **0.330** | **0.381** |
| | 192 | 0.334 | 0.376 | **0.331** | **0.375** | 0.397 | 0.428 | **0.335** | **0.378** | 0.442 | 0.435 | **0.344** | **0.383** | 0.446 | 0.438 | **0.435** | **0.430** |
| | 336 | 0.357 | 0.399 | **0.354** | **0.397** | 0.389 | 0.426 | **0.326** | **0.398** | 0.456 | 0.461 | **0.370** | **0.406** | 0.493 | 0.471 | **0.477** | **0.448** |
| | 720 | 0.388 | 0.427 | **0.379** | **0.425** | 0.430 | 0.454 | **0.380** | **0.420** | 0.452 | 0.459 | **0.401** | **0.439** | 0.444 | 0.458 | 0.568 | 0.547 |
| ETTm1 | 96 | 0.308 | 0.347 | **0.300** | **0.343** | 0.679 | 0.544 | **0.305** | **0.347** | 0.691 | 0.541 | **0.311** | **0.356** | 0.421 | 0.437 | **0.336** | **0.375** |
| | 192 | 0.341 | 0.366 | **0.337** | **0.365** | 0.687 | 0.549 | **0.338** | **0.366** | 0.708 | 0.550 | **0.341** | **0.373** | 0.426 | 0.421 | **0.382** | **0.401** |
| | 336 | 0.375 | 0.387 | **0.368** | **0.386** | 0.700 | 0.556 | **0.372** | **0.387** | 0.719 | 0.558 | **0.376** | **0.393** | 0.481 | 0.468 | **0.423** | **0.433** |
| | 720 | 0.427 | 0.416 | **0.418** | **0.414** | 0.718 | 0.569 | **0.427** | **0.416** | 0.742 | 0.572 | **0.426** | **0.420** | 0.540 | 0.495 | **0.510** | **0.476** |
| ETTm2 | 96 | 0.165 | 0.254 | **0.161** | **0.251** | 0.277 | 0.346 | **0.167** | **0.256** | 0.227 | 0.308 | **0.167** | **0.257** | 0.202 | 0.283 | **0.198** | **0.280** |
| | 192 | 0.219 | 0.292 | **0.215** | **0.289** | 0.306 | 0.362 | **0.222** | **0.293** | 0.284 | 0.338 | **0.231** | **0.302** | 0.278 | 0.328 | 0.328 | 0.381 |
| | 336 | 0.272 | 0.327 | **0.267** | **0.325** | 0.341 | 0.380 | **0.277** | **0.329** | 0.339 | 0.370 | **0.289** | **0.338** | 0.336 | 0.364 | **0.330** | **0.354** |
| | 720 | 0.366 | 0.382 | **0.349** | **0.379** | 0.422 | 0.424 | **0.366** | **0.382** | 0.434 | 0.419 | **0.374** | **0.392** | 0.441 | 0.423 | 0.522 | 0.513 |

## D.4 SELECTION OF THE BRANCH AGGREGATION STRATEGY

As illustrated in Figure 2, our proposed WaveTS is constructed upon a multi-branch architecture, wherein each branch corresponds to a specific order of Wavelet derivation. Upon obtaining the learned representations from each branch, the fusion and utilization of these representations presents a nontrivial issue. Consequently, this paper introduces two branch aggregation strategies aimed at identifying the optimal solution based on numerical results. The results, summarized in Table 11, indicate that concatenation along the temporal dimension consistently outperforms concatenation along the channel dimension. These findings demonstrate that the multi-scale patterns excavated by each branch are time-aware, reflecting temporal variations while aligning with the channel independence of the WDT within WaveTS. A similar phenomenon is also observed in experiments on the similar multi-branch DeRiTS (Fan et al., 2014).

Table 11: Comparison of different concatenation strategies. (C) denotes concatenation along the channel dimension, while (T) denotes concatenation along the temporal dimension.

| Models Metrics | | WaveTS(C) MSE | MAE | WaveTS(T) MSE | MAE | DeRiTS(C) MSE | MAE | DeRiTS(T) MSE | MAE |
|---|---|---|---|---|---|---|---|---|---|
| ETTh1 | 96 | 0.409 | 0.423 | **0.370** | **0.393** | 0.625 | 0.531 | **0.447** | **0.452** |
| | 192 | 0.426 | 0.433 | **0.404** | **0.413** | 0.665 | 0.550 | **0.471** | **0.465** |
| | 336 | 0.444 | 0.449 | **0.429** | **0.428** | 0.710 | 0.574 | **0.475** | **0.471** |
| | 720 | 0.456 | 0.468 | **0.436** | **0.454** | 0.730 | 0.608 | **0.469** | **0.486** |
| ETTh2 | 96 | 0.345 | 0.393 | **0.270** | **0.335** | 0.380 | 0.400 | **0.316** | **0.382** |
| | 192 | 0.361 | 0.401 | **0.331** | **0.375** | 0.442 | 0.435 | **0.355** | **0.407** |
| | 336 | 0.362 | 0.407 | **0.354** | **0.397** | 0.456 | 0.461 | **0.386** | **0.429** |
| | 720 | 0.425 | 0.454 | **0.379** | **0.425** | 0.452 | 0.459 | **0.416** | **0.454** |
| ETTm1 | 96 | 0.329 | 0.364 | **0.300** | **0.343** | 0.691 | 0.541 | **0.612** | **0.522** |
| | 192 | 0.255 | 0.314 | **0.337** | **0.365** | 0.708 | 0.550 | **0.471** | **0.465** |
| | 336 | 0.383 | 0.392 | **0.368** | **0.386** | 0.719 | 0.558 | **0.640** | **0.538** |
| | 720 | 0.435 | 0.422 | **0.418** | **0.414** | 0.742 | 0.572 | **0.667** | **0.554** |
| ETTm2 | 96 | 0.175 | 0.265 | **0.161** | **0.251** | 0.257 | 0.338 | **0.248** | **0.334** |
| | 192 | 0.354 | 0.377 | **0.215** | **0.289** | 0.299 | 0.358 | **0.285** | **0.355** |
| | 336 | 0.290 | 0.338 | **0.267** | **0.325** | 0.339 | 0.380 | **0.326** | **0.375** |
| | 720 | 0.416 | 0.411 | **0.349** | **0.379** | 0.439 | 0.429 | **0.415** | **0.423** |
| Exchange | 96 | 0.176 | 0.310 | **0.084** | **0.203** | 0.268 | 0.375 | **0.134** | **0.277** |
| | 192 | 0.305 | 0.414 | **0.175** | **0.296** | 0.359 | 0.447 | **0.238** | **0.365** |
| | 336 | 0.426 | 0.487 | **0.339** | **0.421** | 0.523 | 0.547 | **0.379** | **0.463** |
| | 720 | 1.155 | 0.816 | **0.864** | **0.693** | 1.383 | 0.891 | **0.924** | **0.736** |

### D.5 HYPERPARAMETER SENSITIVITY ANALYSIS ON ADDITIONAL DATASETS

We evaluate the sensitivity of our model to two key hyperparameters, the order $O$ and the scale $S$, on additional datasets: ETTm1, Weather, and ECL. Table 12 reports forecasting performance when varying $O$ (with $S = 5$ fixed), and Table 13 shows the results for different $S$ (with $O = 2$ fixed). We also provide results on even larger $O$ and $S$ in Figure 9. Despite altering these hyperparameters, the changes in forecasting accuracy are relatively minor, indicating that our model maintains robust performance under varying settings.

Table 12: Forecasting performance across horizons, scale is fixed to 5 and order ranges from 1 to 4.

| Horizon | ETTm1 | | | | Weather | | | | ECL | | | |
|---|---|---|---|---|---|---|---|---|---|---|---|---|
| | $O$=1 | $O$=2 | $O$=3 | $O$=4 | $O$=1 | $O$=2 | $O$=3 | $O$=4 | $O$=1 | $O$=2 | $O$=3 | $O$=4 |
| 96 | 0.301 | 0.303 | 0.308 | 0.305 | 0.168 | 0.171 | 0.170 | 0.169 | 0.134 | 0.134 | 0.134 | 0.134 |
| 192 | 0.341 | 0.338 | 0.340 | 0.340 | 0.210 | 0.210 | 0.213 | 0.212 | 0.149 | 0.149 | 0.149 | 0.148 |
| 336 | 0.375 | 0.375 | 0.373 | 0.374 | 0.258 | 0.259 | 0.255 | 0.258 | 0.165 | 0.165 | 0.165 | 0.164 |
| 720 | 0.428 | 0.431 | 0.430 | 0.430 | 0.321 | 0.321 | 0.322 | 0.322 | 0.204 | 0.204 | 0.205 | 0.206 |

Table 13: Forecasting performance across horizons, order is fixed to 2 and scale ranges from 2 to 5.

| Horizon | ETTm1 | | | | Weather | | | | ECL | | | |
|---|---|---|---|---|---|---|---|---|---|---|---|---|
| | $S$=2 | $S$=3 | $S$=4 | $S$=5 | $S$=2 | $S$=3 | $S$=4 | $S$=5 | $S$=2 | $S$=3 | $S$=4 | $S$=5 |
| 96 | 0.305 | 0.306 | 0.303 | 0.303 | 0.167 | 0.169 | 0.169 | 0.171 | 0.134 | 0.134 | 0.134 | 0.134 |
| 192 | 0.342 | 0.344 | 0.338 | 0.338 | 0.212 | 0.211 | 0.214 | 0.210 | 0.149 | 0.149 | 0.149 | 0.149 |
| 336 | 0.375 | 0.374 | 0.375 | 0.375 | 0.258 | 0.257 | 0.257 | 0.259 | 0.165 | 0.165 | 0.165 | 0.165 |
| 720 | 0.430 | 0.432 | 0.429 | 0.431 | 0.322 | 0.320 | 0.322 | 0.321 | 0.204 | 0.204 | 0.204 | 0.204 |

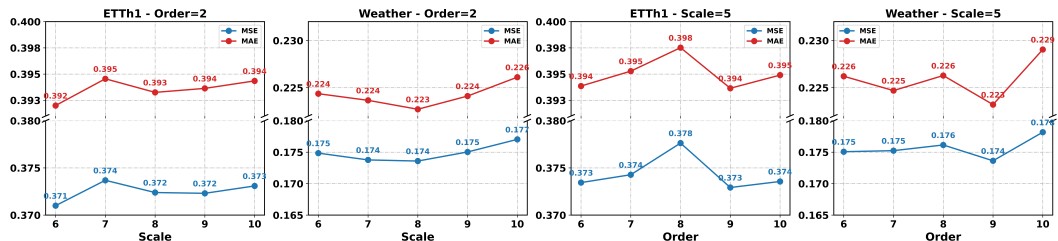

Figure 9: Model performance variations under different scales and orders in the WDT module. Results are collected from forecasting 96 experiments, each using its optimal input length.

### D.6 FAIR AND REPRODUCIBLE HYPERPARAMETER SELECTION

**Lookback Window Length** $L$. We treat $L$ as a tunable hyperparameter because architectures require different effective receptive fields to perform optimally. Enforcing a single $L$ (e.g., 96) conflates equality with fairness and disadvantages models (e.g., DLinear (Zeng et al., 2023)) that benefit from longer histories, while many Transformer variants gain little from longer inputs. For WaveTS, a larger $L$ improves time–frequency fidelity and remains affordable; thus we tune $L$ on the validation set. For baselines, we strictly follow lookbacks from their papers/official code (defaults when unspecified). Evaluating each method under its own validated configuration is widely accepted and improves fairness and reproducibility (Zeng et al., 2023; Yang et al., 2024; Xu et al., 2024).

**WDT Order** $O$ **and Scale** $S$. WaveTS uses the WDT derivative order $O$ (number of branches $N$ in Eq. 3) and scale $S$ (decomposition levels $K$ in Eq. 3). For each dataset and horizon, we pick $(O, S)$ via a tiny, validation-driven grid (optionally with Optuna), typically $O, S \in \{1, \ldots, 5\}$. There is no closed-form rule from data traits to optimal $(O, S)$, so the search is principled, lightweight, and reproducible. Sensitivity analyses in Figure 3 and Figure 9 show performance is flat around the optimum: periodic/stable datasets (e.g., Traffic, Electricity) often prefer smaller $O, S$; more non-stationary, multi-scale datasets benefit from larger $O, S$; longer horizons favor a larger $S$.

### D.7 ADDITIONAL ANALYSIS ON THE ORDER AND THE SCALE

**Contribution of each branch.** As described in Section 3.2.2, after obtaining the multi-branch time-domain representations $\{\mathbf{Z}_n\}_{n=1}^N$ in Eq. 5, we concatenate them along the time dimension and apply a linear projection to obtain the final reconstruction and prediction as in Eq. 6. To make the role of each derivative-order branch explicit, we provide a quantitative analysis of its contribution to the final forecast. Because the final aggregation is **strictly linear**, the projection weight in Eq. 6 can be written as the block matrix $\mathbf{W} = [\mathbf{W}^{(1)}\ \mathbf{W}^{(2)}\ \cdots\ \mathbf{W}^{(N)}]$, where each block $\mathbf{W}^{(n)}$ multiplies the corresponding branch output $\mathbf{Z}_n$. This yields the exact additive decomposition $\mathbf{Y}_t = \sum_{n=1}^N \mathbf{W}^{(n)}\mathbf{Z}_n + \mathbf{b}$, and we define the per-branch contribution as $\mathbf{Y}_t^{(n)} = \mathbf{W}^{(n)}\mathbf{Z}_n$, so that $\mathbf{Y}_t = \sum_{n=1}^N \mathbf{Y}_t^{(n)} + \mathbf{b}$. Since no nonlinearity is applied after concatenation, there are no cross terms and the superposition principle holds. We quantify each branch's impact using the $\ell_2$ energy share $\rho_n^{(2)} = \frac{\|\mathbf{Y}_t^{(n)}\|_2}{\sum_{m=1}^N \|\mathbf{Y}_t^{(m)}\|_2}$, computed on the test set. This incurs no retraining and only a single forward pass, together with a closed-form split of $\mathbf{W}$ into its $N$ column blocks $\{\mathbf{W}^{(n)}\}$. Tables 14 and 15 report $\rho_n^{(2)}$ for WaveTS with four branches ($O = 4$) on the Traffic and ETTm2 datasets across horizons $\{96, 192, 336, 720\}$. Lower-order branches dominate on strongly periodic data (Traffic), whereas higher-order branches contribute more on more nonstationary data (ETTm2), which is consistent with our hyperparameter selection rationale. Boldface marks the largest contribution.

Table 14: $\ell_2$ energy share on Traffic (larger indicates a greater contribution).

|  | Traffic_96 | Traffic_192 | Traffic_336 | Traffic_720 |
| --- | --- | --- | --- | --- |
| Branch 1 ($O = 1$) | **0.301543** | **0.299975** | **0.316984** | 0.264803 |
| Branch 2 ($O = 2$) | 0.281398 | 0.250889 | 0.254451 | **0.276797** |
| Branch 3 ($O = 3$) | 0.217832 | 0.244350 | 0.242937 | 0.219048 |
| Branch 4 ($O = 4$) | 0.199227 | 0.204786 | 0.185628 | 0.249352 |

Table 15: $\ell_2$ energy share on ETTm2 (larger indicates a greater contribution).

|  | ETTm2_96 | ETTm2_192 | ETTm2_336 | ETTm2 _720 |
| --- | --- | --- | --- | --- |
| Branch 1 ($O = 1$) | 0.242765 | 0.244528 | 0.237057 | 0.250936 |
| Branch 2 ($O = 2$) | 0.242479 | 0.228037 | 0.254821 | 0.222338 |
| Branch 3 ($O = 3$) | 0.251197 | **0.264088** | 0.230385 | **0.264954** |
| Branch 4 ($O = 4$) | **0.263559** | 0.263347 | **0.277738** | 0.261772 |

**Which orders and scales suit which datasets?** Complementing the contribution analysis above, we examine which derivative order $O$ and wavelet scale $S$ are suitable for different datasets and horizons. In our setting, $O$ equals the WDT derivative order and $S$ is the number of wavelet decomposition levels. For each dataset and each forecasting horizon, we select $(O, S)$ on the validation set via a small grid (optionally accelerated by Optuna), typically $O, S \in \{1, \ldots, 5\}$. We adopt this protocol because there is no closed form mapping from dataset characteristics such as periodicity and nonstationarity, together with horizon length, to the optimal $(O, S)$. The search space is tiny, so the procedure is negligible in cost and reproducible; moreover, our sensitivity analyses show performance is relatively flat near the optimum, indicating the search chooses among near equivalent settings rather than overfitting. Empirically, the validated choices in Tables 16 and 17 follow a consistent pattern: strongly periodic datasets such as Traffic and Electricity tend to prefer smaller $O$ and $S$, while datasets with more pronounced nonstationarity or multi scale fluctuations such as ETTm1 and ETTm2 often benefit from larger $O$ and $S$. Longer horizons usually favor a slightly larger $S$ to stabilize low frequency forecasting.

### D.8 SELECTION OF THE WAVELET BASIS FUNCTION

Selecting an appropriate Wavelet basis can be non-trivial, as each candidate has unique filter lengths and boundary effects. In our experiments (input = 360, horizon = 96), results in Table 18 show minimal performance differences among three short-length Wavelet bases: `bior1.1`, `rbio1.1`,

Table 16: Traffic and Electricity.

| Traffic | | | Electricity | | |
|---|---|---|---|---|---|
| Horizon | $O$ | $S$ | Horizon | $O$ | $S$ |
| 96 | 2 | 3 | 96 | 2 | 2 |
| 192 | 3 | 2 | 192 | 3 | 2 |
| 336 | 3 | 4 | 336 | 4 | 3 |
| 720 | 2 | 4 | 720 | 3 | 5 |

Table 17: ETTm1 and ETTm2.

| ETTm1 | | | ETTm2 | | |
|---|---|---|---|---|---|
| Horizon | $O$ | $S$ | Horizon | $O$ | $S$ |
| 96 | 3 | 2 | 96 | 2 | 2 |
| 192 | 3 | 3 | 192 | 4 | 3 |
| 336 | 4 | 5 | 336 | 4 | 4 |
| 720 | 5 | 5 | 720 | 4 | 5 |

and `db1` (Haar). We focus on short filters because they provide consistent dyadic decompositions, ensuring each level halves the temporal dimension without introducing significant boundary overlap. Longer Wavelet filters (e.g., `db2`, `coif5`) can lead to overlapping boundaries, mismatch in LL/LH decomposition dimensions, and increased model errors. Moreover, `db1` is a real-valued Wavelet basis (often referred to as the Haar Wavelet), which avoids additional complexities arising from complex-valued Wavelets. Hence, we select `db1` (Haar) in practice, balancing simplicity, fast computation, and reliable decomposition properties.

Table 18: Forecasting performance of different Wavelet bases on eight datasets .

| Wavelet | ETTh1 | | ETTh2 | | ETTm1 | | ETTm2 | |
|---|---|---|---|---|---|---|---|---|
| | MSE | MAE | MSE | MAE | MSE | MAE | MSE | MAE |
| bior1.1 | 0.371 | 0.393 | 0.275 | 0.337 | 0.305 | 0.348 | 0.166 | 0.256 |
| db1 | 0.373 | 0.394 | 0.273 | 0.336 | 0.304 | 0.347 | 0.166 | 0.256 |
| rbio1.1 | 0.372 | 0.393 | 0.276 | 0.338 | 0.303 | 0.346 | 0.165 | 0.255 |

| Wavelet | Exchange | | ECL | | Traffic | | Weather | |
|---|---|---|---|---|---|---|---|---|
| | MSE | MAE | MSE | MAE | MSE | MAE | MSE | MAE |
| bior1.1 | 0.087 | 0.206 | 0.133 | 0.229 | 0.383 | 0.267 | 0.169 | 0.224 |
| db1 | 0.089 | 0.208 | 0.134 | 0.231 | 0.384 | 0.269 | 0.171 | 0.226 |
| rbio1.1 | 0.089 | 0.208 | 0.134 | 0.230 | 0.384 | 0.269 | 0.171 | 0.226 |

## D.9 ABLATION STUDY

The WaveTS framework is centered on the Wavelet Derivative Transform (WDT), which captures multi scale and time aware structure by separating low frequency trends and high frequency fluctuations while preserving exact invertibility. To isolate the effect of the front end transform, we design two ablated models, FourierTS and WaveletTS, by substituting the WDT in WaveTS with the Discrete Fourier Transform (DFT) and the Discrete Wavelet Transform (DWT), respectively, while keeping all other components and training settings identical (normalization, Frequency Refinement Units, linear projection, and losses). The ablation architectures are illustrated in Figure 10. Results on four ETT datasets and four PEMS datasets are summarized in Table 19, showing that WDT consistently improves long horizon forecasting across metrics and horizons.

**Ablation design and pipelines.** Let $\mathbf{X} \in \mathbb{R}^{L \times C}$ be the normalized input and $\hat{\mathbf{Y}} \in \mathbb{R}^{(L+F) \times C}$ the de normalized output; let $\Pi$ denote the final linear projection along the time dimension. **FourierTS**: apply a DFT to obtain the complex spectrum $\mathbf{H} = \mathcal{F}(X)$ (implemented as real and imaginary channels), refine coefficients $\widehat{\mathbf{H}} = \Phi(\mathbf{H})$ with the same refinement units as in WaveTS, invert $\tilde{z} = \mathcal{F}^{-1}(\widehat{\mathbf{H}})$, and predict $\hat{\mathbf{Y}} = \Pi(\tilde{z})$. **WaveletTS**: apply a $K$ level DWT $(\mathbf{LH}_K, LL_K) = \mathcal{W}_K(\mathbf{X})$ with $\mathbf{LH}_K = \{LH_1, \ldots, LH_K\}$; refine per level $\widehat{LL}_K = \Phi_K(LL_K)$ and $\widehat{\mathbf{LH}}_K = \Phi_K(\mathbf{LH}_K)$; invert $\tilde{z} = \mathcal{W}_K^{-1}(\widehat{LL}_K, \widehat{\mathbf{LH}}_K)$; and predict $\hat{\mathbf{Y}} = \Pi(\tilde{z})$. In contrast, WaveTS uses multi order branching with derivative aware WDT per branch before reconstruction and aggregation. Thus, the ablations differ from WaveTS in two principled aspects: (i) absence of derivative amplification and multi order branching, and (ii) spectral structure, where FourierTS models a single global spectrum while WaveletTS and WaveTS operate on band localized coefficients, with only WaveTS coupling band localization and order wise derivative emphasis.

Table 19: Ablation of the transform front end on four ETT datasets in long horizon forecasting. "I/O" indicates lookback window size and forecasting length. FourierTS and WaveletTS replace WDT with DFT and DWT in WaveTS, respectively.

| Dataset | ETTh1 | | | | | | | | ETTh2 | | | | | | | |
|---|---|---|---|---|---|---|---|---|---|---|---|---|---|---|---|---|
| I/O | 336/96 | | 336/192 | | 336/336 | | 336/720 | | 336/96 | | 336/192 | | 336/336 | | 336/720 | |
| Metrics | MSE | MAE | MSE | MAE | MSE | MAE | MSE | MAE | MSE | MAE | MSE | MAE | MSE | MAE | MSE | MAE |
| FourierTS | 0.374 | 0.396 | 0.407 | 0.416 | 0.433 | 0.432 | 0.447 | 0.461 | 0.273 | 0.339 | 0.341 | 0.384 | 0.368 | 0.411 | 0.396 | 0.434 |
| WaveletTS | 0.374 | 0.395 | 0.408 | 0.416 | 0.432 | 0.430 | 0.447 | 0.459 | 0.274 | 0.339 | 0.339 | 0.378 | 0.362 | 0.402 | 0.393 | 0.431 |
| **WaveTS** | **0.370** | **0.393** | **0.404** | **0.413** | **0.429** | **0.428** | **0.436** | **0.454** | **0.270** | **0.335** | **0.331** | **0.375** | **0.354** | **0.397** | **0.379** | **0.425** |
| Dataset | ETTm1 | | | | | | | | ETTm2 | | | | | | | |
| I/O | 336/96 | | 336/192 | | 336/336 | | 336/720 | | 336/96 | | 336/192 | | 336/336 | | 336/720 | |
| Metrics | MSE | MAE | MSE | MAE | MSE | MAE | MSE | MAE | MSE | MAE | MSE | MAE | MSE | MAE | MSE | MAE |
| FourierTS | 0.305 | 0.347 | 0.340 | 0.369 | 0.371 | 0.389 | 0.428 | 0.417 | 0.166 | 0.256 | 0.221 | 0.294 | 0.274 | 0.329 | 0.376 | 0.391 |
| WaveletTS | 0.305 | 0.347 | 0.341 | 0.369 | 0.372 | 0.389 | 0.430 | 0.418 | 0.165 | 0.255 | 0.219 | 0.292 | 0.274 | 0.329 | 0.369 | 0.383 |
| **WaveTS** | **0.300** | **0.343** | **0.337** | **0.365** | **0.368** | **0.386** | **0.418** | **0.414** | **0.161** | **0.251** | **0.215** | **0.289** | **0.267** | **0.325** | **0.349** | **0.379** |
| Dataset | PEMS03 | | | | PEMS04 | | | | PEMS07 | | | | PEMS08 | | | |
| I/O | 96/12 | | 96/24 | | 96/12 | | 96/24 | | 96/12 | | 96/24 | | 96/12 | | 96/24 | |
| Metrics | MSE | MAE | MSE | MAE | MSE | MAE | MSE | MAE | MSE | MAE | MSE | MAE | MSE | MAE | MSE | MAE |
| FourierTS | 0.126 | 0.247 | 0.241 | 0.342 | 0.137 | 0.260 | 0.253 | 0.358 | 0.116 | 0.242 | 0.239 | 0.348 | 0.132 | 0.258 | 0.242 | 0.349 |
| WaveletTS | 0.080 | 0.181 | 0.113 | 0.212 | 0.096 | 0.199 | 0.127 | 0.232 | 0.075 | 0.179 | 0.109 | 0.214 | 0.096 | 0.196 | 0.143 | 0.233 |
| **WaveTS** | **0.075** | **0.178** | **0.107** | **0.208** | **0.091** | **0.196** | **0.122** | **0.228** | **0.071** | **0.174** | **0.103** | **0.209** | **0.091** | **0.193** | **0.138** | **0.231** |

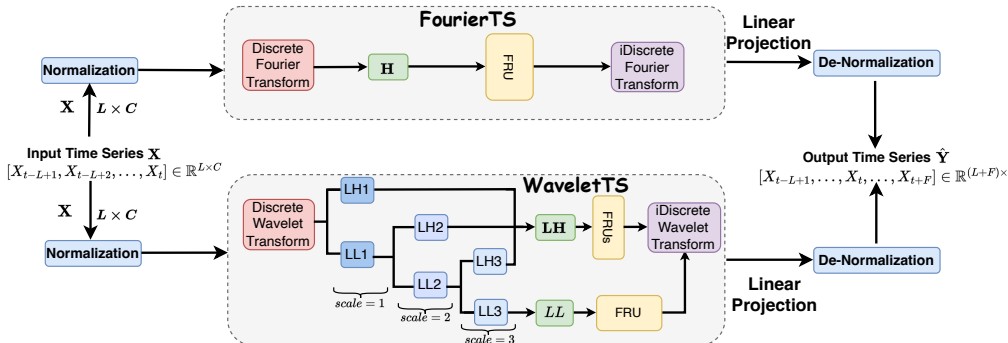

Figure 10: FourierTS and WaveletTS for the ablation, where DFT and DWT replace WDT in WaveTS.

Across all datasets and horizons in Table 19, WaveTS surpasses both FourierTS and WaveletTS. The gains are most pronounced at longer horizons, where derivative aware multi order branching provides richer low frequency stabilization and high frequency discrimination than a global Fourier spectrum or a derivative free wavelet stack. These findings support the design choice of coupling band localization with derivative emphasis in WaveTS.

# E    VISUALIZATIONS

## E.1    VISUALIZATIONS OF THE FORECASTING PERFORMANCE

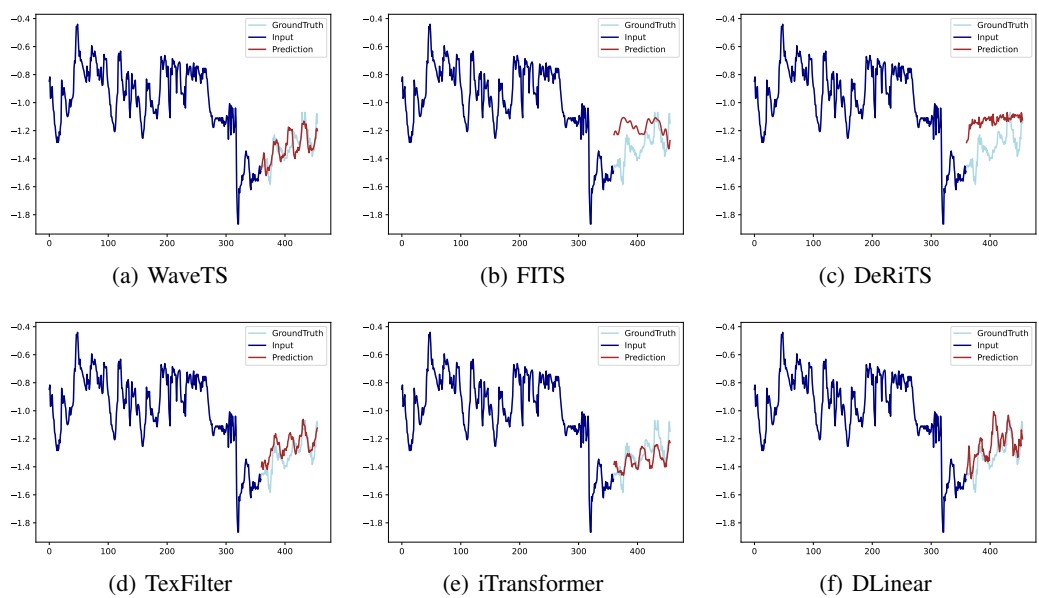

Figure 11: Visualizations of the results on the ETTh1 dataset with lookback window size of 360 and forecasting length of 96.

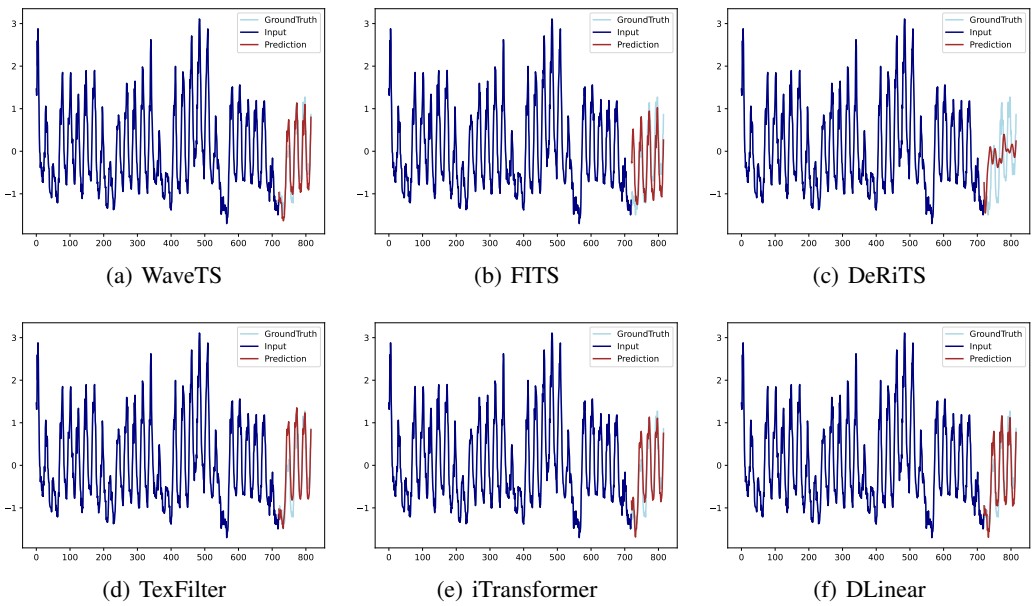

Figure 12: Visualizations of the results on the Electricity dataset with lookback window size of 720 and forecasting length of 96.

## E.2 ADDITIONAL VISUALIZATIONS OF THE COEFFICIENTS IN THE WDT

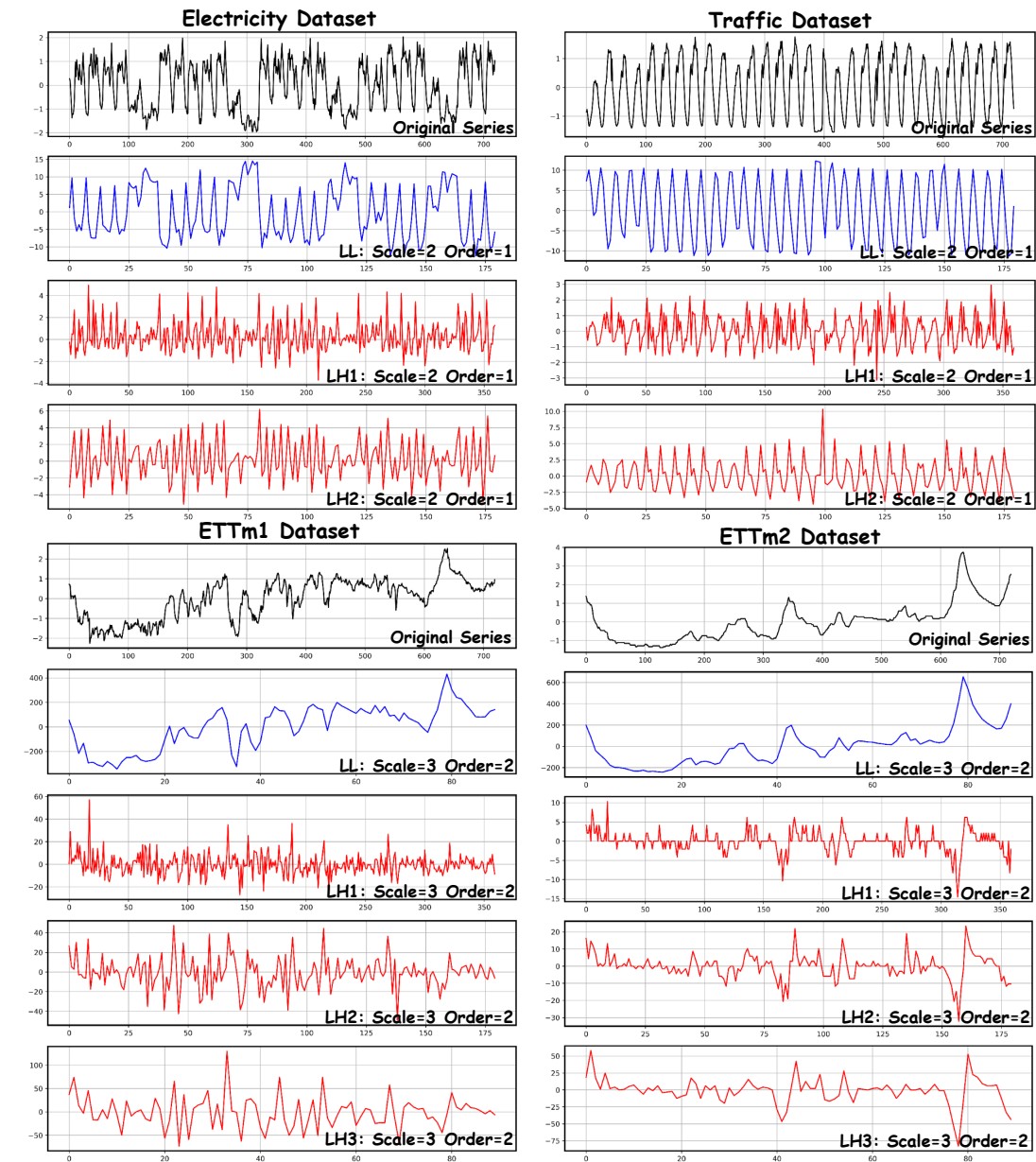

Figure 13: Additional visualizations of the Approximation Coefficients (LL) and Detail Coefficients (LH) in the WDT on four different datasets. We select the "OT" variate with the input length is 720 for each dataset.

Derivative of the sequence is essential for identifying more semantically rich and robust representations. As illustrated in Figure 13, LL reveals macro temporal patterns, providing insight into the overall trend, while LH captures detailed fluctuations and complex irregular micro patterns at various scales. Prior research on frequency derivative learning indicates that derivation yields more stationary frequency patterns, as depicted in Figure 5 in (Fan et al., 2014). However, the conversion of the original sequence into stationary representations may neglect to capture the overall trend, the most fundamental and informative temporal pattern indicating nonstationarity, thereby resulting in ineffective modeling. As demonstrated in Figure 13, WaveTS effectively addresses this problem by decomposing the original signal into LL and LH coefficients, wherein the former encapsulates trend and periodicity, while the latter captures more stationary characteristics. Consequently, WaveTS is capable of modeling hierarchical and robust frequency features simultaneously, which significantly enhances forecast performance.

### E.3 ADDITIONAL VISUALIZATIONS OF THE TIME-FREQUENCY SCALOGRAM

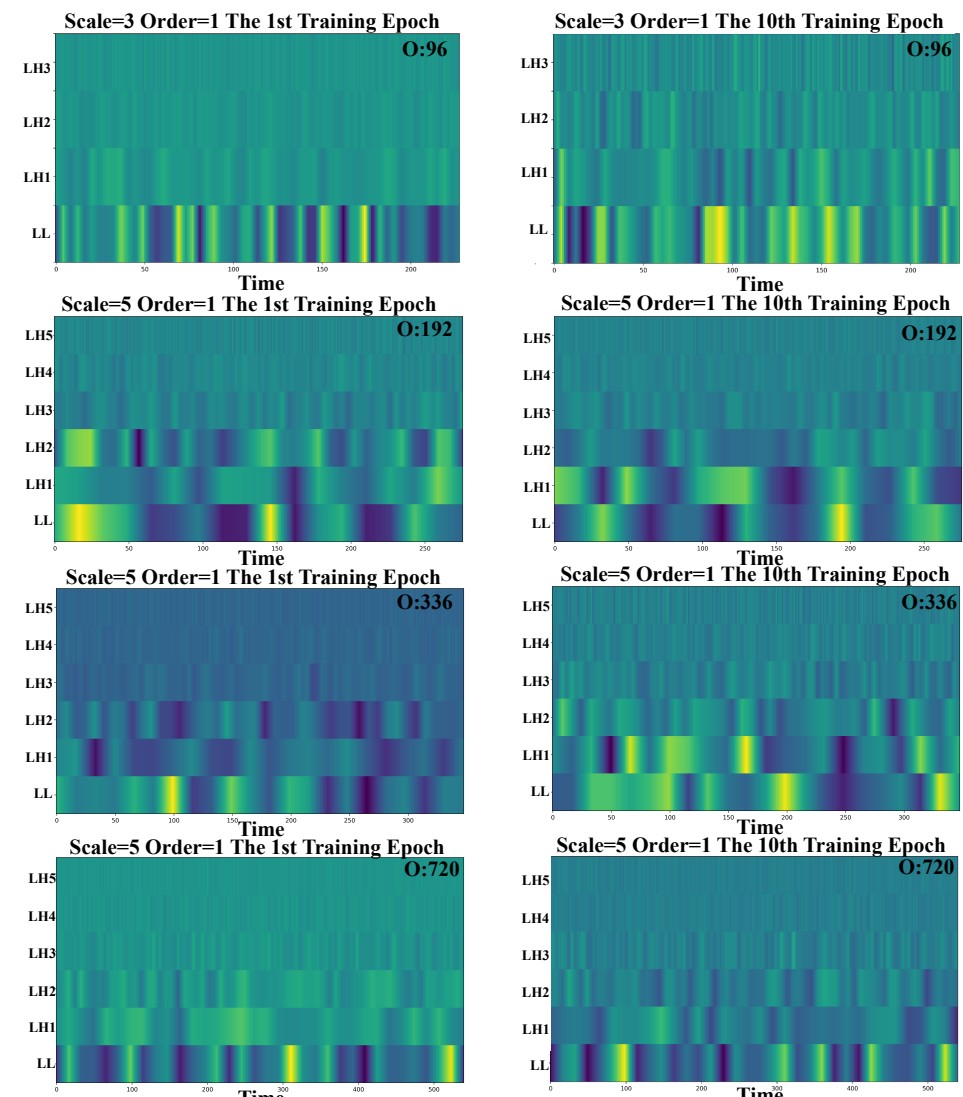

Figure 14: Scalograms are obtained from the training process on the ETTm1 dataset. The input length is 360, with the output length ranging from 96 to 720. We set scale=3 for forecasting a horizon of 96 and adjust the scale for other forecasting horizons based on their best performance. For each forecasting horizon, we visualize the scalogram from the first and the last training epochs. The color bar here is identical to that in Figure 6.

## F LIMITATIONS

While WaveTS achieves state-of-the-art accuracy and efficiency on a diverse collection of regularly-sampled benchmarks, two avenues for improvement remain. First, our empirical study is confined to fixed-interval datasets; extending the model to irregularly-sampled or event-driven series would clarify its robustness in real-world monitoring scenarios.

## G THE USE OF LARGE LANGUAGE MODELS (LLMs)

We employ LLMs solely to correct grammatical errors and to ensure the writing is fluent, accurate, and coherent. The authors take full responsibility for all content generated with LLM assistance.

