# OpenReview forum: "Multi-Order Wavelet Derivative Transform for Deep Time Series Forecasting"
_ICLR.cc/2026/Conference — Submitted to ICLR 2026_

### Official Review · Reviewer_5xKq · 2025-10-25

**Soundness:** 3
**Presentation:** 1
**Contribution:** 3
**Rating:** 4
**Confidence:** 5

**Summary:**

The paper proposes the Wavelet Derivative Transform (WDT) based on the Wavelet Transform. WDT operates on the derivative of the time series, selectively amplifying rate-of-change cues and revealing abrupt regime shifts that are especially informative for time series modeling. The paper provides extensive theoretical analysis of WDT, including proofs related to the inverse transform, differentiation, and other properties.

**Strengths:**

- The paper provides numerous proofs, which appear meaningful.

- Developing WDT and IWDT seems non-trivial, although the paper does not clearly articulate the specific challenges involved.

- The experimental results in the paper validate the effectiveness of the proposed model.

**Weaknesses:**

The introduction is poorly written. It fails to convey why the paper is important, even though the proofs presented later seem significant.

1. The paper does not effectively connect its theoretical proofs with the broader discussion. The introduction completely omits these key elements—rendering it, in a sense, largely vacuous. It fails to clarify the difficulty of the problem being addressed, the significance of the contribution, or the approach taken to solve it. If the proofs are indeed as important as I understand them to be, the authors should have summarized their challenges and implications right in the introduction. Otherwise, readers less familiar with the field will struggle to grasp the significance of these results, even if they read the conclusions of the proofs.

2. When X is a discrete sequence, is directly computing its derivative inherently difficult? Do WDT and IWDT (Inverse Wavelet Derivative Transform) address this challenge?

3. Is it difficult to construct a basis that possesses the inverse transform property? What is the practical or theoretical significance of having an inverse transform?

4. What is the meaning (or required properties) of wavelet basis functions? Does WDT satisfy these properties? Does WDT offer additional desirable properties beyond those of conventional wavelet transforms?

5. The paper employs $\psi^{(n)} $ (the n-th derivative of the mother wavelet $\psi$ ), yet for many mother wavelets $\psi$ , computing such derivatives is nontrivial. How are these derivatives obtained in practice? This constitutes a significant technical challenge that the paper should explicitly address—either through explanation or formal justification.

**Questions:**

Please address Weaknesses 1–5.

If Weakness 5 is adequately answered, I will raise my rating to 6.
If the responses to the other weaknesses are also satisfactory, I will consider raising my rating to 8.

---

> ### Author Response · Authors · 2025-11-22
> **Response to Weakness 1 and Weakness 2**
>
> ## **Response to Reviewer 5xKq**
>
> We sincerely thank Reviewer 5xKq for the careful and technically detailed evaluation of our work. Below we respond to your concerns point by point and have revised the related sections in the paper. We are willing to address any remaining issues in the subsequent rebuttal. We also kindly invite you to download the revised PDF, in which several modifications have been included.
>
> ---
>
> **Q1: The paper does not effectively connect its theoretical proofs with the broader discussion. The introduction completely omits these key elements—rendering it, in a sense, largely vacuous. It fails to clarify the difficulty of the problem being addressed, the significance of the contribution, or the approach taken to solve it. If the proofs are indeed as important as I understand them to be, the authors should have summarized their challenges and implications right in the introduction. Otherwise, readers less familiar with the field will struggle to grasp the significance of these results, even if they read the conclusions of the proofs.**
>
> **A1:**
> Thank you for pointing this out. In the revised manuscript, we explicitly connect the theoretical proofs with the main narrative in the Introduction, and all changes are marked in **blue**:
>
> * As suggested, in the revised Introduction section, we have added explanations on what our main theorems prove, why constructing an invertible multi-order wavelet derivative transform is non-trivial, and why these results matter for frequency-domain forecasting.
>
> * We add a separate contribution item summarizing our theoretical results (invertibility, stability/energy behavior, and the derivative interpretation of WDT).
>
> * We slightly rephras the end of the Introduction to make clear that these guarantees underpin the design and reliability of WaveTS.
>
> We hope these concise additions make the difficulty and significance of our theoretical results more transparent to readers.
>
> ---
>
> **Q2.1: When X is a discrete sequence, is directly computing its derivative inherently difficult?**
>
> **A2.1:**
> Yes, directly computing derivatives of discrete sequences is inherently difficult. As stated in Section 3.1 (lines 176-178, revised version), time-domain differentiation suffers from three critical issues: finite differences amplify high-frequency noise, each differentiation reduces the usable sequence length, and complications arise at sequence boundaries. These issues make direct numerical differentiation fragile and unsuitable for robust time series modeling.
>
> **Q2.2: Do WDT and IWDT address this challenge?**
>
> **A2.2:**
> Yes, WDT is specifically designed to address this challenge. Recognizing the difficulties of direct numerical differentiation, we strategically shift from time-domain numerical differentiation to spectral-domain differentiation. The key insight (Definition 1, Sec 3.1) is that we convolve $X(t)$ with the $n$-th derivative of the mother wavelet rather than differentiating $X(t)$ directly: $\text{WDT}^{(n)} _ {k,j} = (-1)^n 2^{nk} \tilde{W}^{(n)} _ {k,j}$. As stated in lines 178-180 (revised version), this spectral approach "keeps the transform perfectly invertible, suppresses numerical noise and preserves scale alignment." The iWDT (Definition 2, Eq. (2)) guarantees lossless reconstruction, with rigorous proofs in Appendix B.1-B.2. This completely avoids the pitfalls of numerical differentiation while capturing multi-scale derivative information for modeling non-stationary dynamics.

---

> ### Author Response · Authors · 2025-11-22
> **Response to Weakness 3 and Weakness 4**
>
> **Q3.1: Is it difficult to construct a basis that possesses the inverse transform property?**
>
> **A3.1:**
> No. In our work we do not design a new wavelet basis. We directly use the standard orthonormal DWT basis with perfect reconstruction (Eq. (1)), and define WDT as a **coefficient-wise linear rescaling operator** $D^{(n)}$ on these DWT coefficients (Definition 1, Eq. (3)). Since $D^{(n)}$ is diagonal with non-zero scale factors, it is trivially invertible, and iWDT is simply “inverse rescaling + inverse DWT” (Definition 2, Eq. (2)). Thus, the inverse-transform property comes essentially for free from the underlying DWT basis, without any complicated basis construction.
>
> **Q3.2: What is the practical or theoretical significance of having an inverse transform?**
>
> **A3.2:**
> The existence of iWDT (Definition 2, Eq. (2)) means that our multi-order wavelet derivative representation is lossless, applying iWDT to WDT coefficients exactly recovers the original signal $X(t)$. Theoretically, WDT is therefore a linear, injective reparameterization of the input, adding derivative-aware structure while preserving all information from the time domain. Practically, because we build on an orthonormal DWT (Eq. (1)), the transform is numerically stable and energy-preserving, and any learned representation in the WDT domain can be mapped back to the time domain for interpretation and reconstruction.
>
> ---
>
> **Q4.1: What is the meaning (or required properties) of wavelet basis functions?**
>
> **A4.1:**
> In our paper, "wavelet basis functions" are the usual dilated-translated copies of an **admissible mother wavelet** $\psi$ ([1,2]), and their derivatives $\psi^{(n)}$, as formally defined in Appendix B.1 (Eqs. (10)-(12)). The required properties are exactly the classical ones ([1,2,3]): (1) $\psi$ is admissible with a well-defined wavelet transform and zero mean; (2) $\psi$ has sufficient decay and vanishing moments so that differentiation and wavelet analysis can be interchanged; (3) the associated (discrete) filter bank generates an orthonormal (or Riesz) basis of $\ell^2$. In practice, we instantiate $\psi$ with an orthonormal Daubechies wavelet (db1 by default) in Sec. 4.1 and Appendix D.8, so the implementation strictly adheres to these standard wavelet-basis requirements.
>
> **Q4.2: Does WDT satisfy these properties?**
>
> **A4.2:**
> Yes. WDT is not a new basis; it is a **diagonal rescaling of coefficients in the derivative wavelet domain** built on the same admissible basis. Appendix B.1 shows that the coefficients associated with the $n$-th derivative basis $\psi^{(n)} _ {k,j}$ are related to the classical wavelet transform of $X^{(n)}$ via known scale–order factors. In Sec. 3.1, Definition 1 defines WDT by applying exactly these factors to standard DWT coefficients, and Definition 2 shows that reconstruction uses the original wavelet basis $\psi _ {k,j}$. Thus WDT **inherits completeness, admissibility, and energy stability** directly from the underlying wavelet transform (Theorem B.2), while remaining a bounded, linear, and invertible operator on the same Hilbert space ($L^2(\mathbb{R})$).
>
> **Q4.3: Does WDT offer additional desirable properties beyond those of conventional wavelet transforms?**
>
> **A4.3:**
> We formally prove the energy conservation property of WDT in Theorem B.2 (Appendix B.1). This property is crucial because it guarantees that no energy is lost when applying WDT and iWDT within the proposed WaveTS framework. It is also worth noting that the FRU in Eq. (4) operates on the entire frequency spectrum and fully leverages all information contained in the WDT coefficients. As a result, WaveTS simultaneously avoids both energy loss and information loss, in contrast to prior methods such as FITS and DeRiTS, where frequency-domain filters explicitly discard part of the spectrum, resulting in inevitable information loss. We do not yet prove additional properties of WDT; thank you for raising this insightful question, which motivates us to further explore more general and fundamental properties for frequency derivative learning in future work.
>
> **References:**
> * [1] Mallat, S. A Wavelet Tour of Signal Processing (3rd ed.). Academic Press, 2008.
> * [2] Daubechies, I. *Ten Lectures on Wavelets.* SIAM, 1992.
> * [3] Strang, G., & Nguyen, T. *Wavelets and Filter Banks.* Wellesley–Cambridge Press, 1996.

---

> ### Author Response · Authors · 2025-11-22
> **Response to Weakness 5, 1/2**
>
> **Q5: The paper employs $\psi^{(n)} $ (the n-th derivative of the mother wavelet $\psi$ ), yet for many mother wavelets $\psi$ , computing such derivatives is nontrivial. How are these derivatives obtained in practice? This constitutes a significant technical challenge that the paper should explicitly address—either through explanation or formal justification.**
>
> **A5:**
> As clarified in this revised version of the paper (Lines 901-909), we do not numerically differentiate $\psi$ in practice. Instead, we implement WDT as **"one standard WT + scale-wise rescaling"**: We perform a single standard DWT with $\psi$, followed by scale- and order-dependent coefficient rescaling. We justify this implementation below.
>
>
> **1. Motivation: Wavelets Function as Multiscale Differential Operators**
>
> Instead of explicitly constructing the derivative filters $\psi^{(n)}$, we leverage the fundamental property of wavelets as **Multiscale Differential Operators**. According to Mallat [1] (Theorem 6.2), a wavelet $\psi$ with $n$ vanishing moments is mathematically equivalent to the $n$-th derivative of a smoothing function $\theta$ (i.e., $\psi = (-1)^n \frac{d^n \theta}{dt^n}$). Consequently, the Continuous Wavelet Transform $W_X(u,s)$:
> $W _ X(u,s) = s^n \frac{d^n}{du^n} (X * \theta _ s)(u).$
> This theorem shows that standard wavelet coefficients $W _ X(u,s)$ **already encode** the derivative information $\frac{d^n}{du^n} (X * \theta _ s)(u)$, requiring only a magnitude correction with the scale-dependent gain $s^n$.
> In other words,
> $$\frac{d^n}{du^n} (X * \theta_s)(u) = s^{-n} W_X(u,s). \tag{1}$$
>
> Analogous to (1), we strictly define the coefficients for the discrete case. Consistent with the vectorized notation $\text{WDT}_k^{(n)}$ in Eq. (8) in our paper, here we explicitly express the relationship for individual coefficients at scale $k$ and translation $j$:
> $ \mathrm{WDT}^{(n)} _ {k,j} = \alpha _ {n,k} W _ X(k,j),$
> where $W _ X(k,j)$ is the standard DWT of $X(t)$ at scale index $k$ and position $j$.
> Here, $\alpha _ {n,k} = (-1)^n 2^{nk}$ is the fixed scaling factor that bridges the continuous theory with our discrete implementation.
>
> Guided by this insight, retrieving the true derivative theoretically requires eliminating this $s^n$ factor. To implement this in the discrete domain (DWT), we adopt an efficient approximation that bridges the gap between the continuous theory and discrete practice. We first perform a single standard DWT of $X(t)$ to obtain coefficients $W_X(k,j)$. Then, acknowledging that the continuous scale $s$ corresponds to dyadic steps $2^k$ in the DWT, we implement the computation of the "$n$-th derivative" coefficients via scalar rescaling: $\mathrm{WDT}^{(n)} _ {k,j} = \alpha _ {n,k} W _ X(k,j)$.
> Here, $\alpha _ {n,k} = (-1)^n 2^{nk}$ is a fixed scaling factor. This value represents the discrete realization of the theoretical inverse gain $s^{-n}$ (consistent with the discrete term $2^{nk}$ derived in Eq. (8) in our paper by substituting the dyadic scale $s=2^k$). By applying this specific $\alpha _ {n,k}$, we effectively invert the scale attenuation $s^n$, thereby recovering the correct magnitude of the differential operator. Crucially, this approach provides a computationally efficient approximation of derivative-aware wavelet analysis (as it only requires standard DWT, which takes $O(T)$ time where $T$ is the time series length). Therefore, our implementation is a theoretically sound realization of the multiscale differential operator that bypasses the need for explicit $\psi^{(n)}$ construction. We acknowledge that due to the discretization gap between the continuous theory (CWT) and our discrete code (DWT), the calculated WDT serves as an approximation of the true continuous derivative (i.e., $\text{Calculated WDT} \approx \text{True Continuous Derivative}$). We further discuss the numerical validity of this approximation in the following section.

---

> ### Author Response · Authors · 2025-11-22
> **Response to Weakness 5, 2/2**
>
> **2. Numerical Validity: db1 Implementation as a Stable Finite Difference Operator**
>
> In our implementation, we unify our choice of mother wavelet to db1 (Line 355). This selection is strategic: unlike other wavelets where the derivative relationship is implicit, db1 offers an explicit numerical equivalence to finite difference operators, rendering our “standard WT + scale-wise rescaling” approach mathematically rigorous.
>
> Specifically, the db1 high-pass analysis filter is $G(z) = \tfrac{1}{\sqrt{2}}(1 - z^{-1})$, which corresponds exactly to the discrete first-difference operator. Consequently, for higher-order derivatives ($n > 1$), applying the iterative filtering in the standard DWT with db1 effectively implements the **binomial difference operator** (denoted as $\Delta^n$), defined in the $z$-domain by the binomial expansion:
> $(1 - z^{-1})^n = \sum _ {m=0}^{n} \binom{n}{m} (-1)^m z^{-m}$. This operator is the standard method for numerically approximating the $n$-th derivative $d^n/dt^n$ in the discrete domain [4]. This equivalence provides a theoretical guarantee for our implementation. From a numerical analysis perspective, we use this discrete difference operator $\Delta^n$ to approximate the true continuous derivative $X^{(n)}(t)$. This finite difference approximation is guaranteed to converge to the true derivative with an error bound of $O(h)$ (derived from the Taylor series expansion: $\frac{\Delta^n X(t)}{h^n}=X^{(n)}(t)+O(h)$). Therefore, by unifying on db1, we ensure that our efficient implementation is not a rough approximation, but serves as a stable discrete realization of a Multiscale Finite Difference Method. This justifies our practical choice of avoiding explicit filter construction in favor of the robust db1-based rescaling.
>
> ---
>
> **To conclude**, the **"one standard WT + scale-wise rescaling"** strategy is not merely an efficient approximation, but a mathematically rigorous solution to implement the n-th derivative of the mother wavelet. All implementation details are available in our code repository: (`https://anonymous.4open.science/r/WaveTS-4review/models/WaveTS.py`).
>
> **References:**
> * [1] Mallat, S. A Wavelet Tour of Signal Processing (3rd ed.). Academic Press, 2008.
> * [2] Daubechies, I. Ten Lectures on Wavelets. SIAM, 1992.
> * [3] Strang, G., Nguyen, T. Wavelets and Filter Banks. Wellesley–Cambridge Press, 1996.
> * [4] LeVeque, R. J. Finite Difference Methods for Ordinary and Partial Differential Equations. SIAM, 2007.

---

> > ### Comment · Reviewer_5xKq · 2025-11-25
> > **Response**
> >
> > Thank you for your response. I'm not sure if my understanding is correct: Although the WDT and IWDT proposed in this paper are based on existing wavelets, they constitute a different set of functions from the perspective of the "mother wavelet."
> > Furthermore, WDT and IWDT satisfy the necessary conditions for wavelets and are capable of extracting derivative features.
> >
> > The paper also has some limitations. For instance, in my view, designing a basis that satisfies numerous stringent properties is not straightforward; therefore, it would be preferable to provide more intuitive explanations of what these properties signify (though this may indeed be challenging).
> >
> > Additionally, the paper would benefit from demonstrating its applicability to wavelet bases other than db1, as the db1 (Haar) basis appears somewhat too simplistic.
> >
> > I hope the authors could elaborate further on these points in the manuscript—additional discussion would likely enhance the paper’s clarity and impact. I appreciate the authors’ thoughtful reply, and I will raise my score accordingly.

---

> > > ### Author Response · Authors · 2025-11-26
> > >
> > > ### **We sincerely appreciate your response and are happy to address your additional comments.**
> > >
> > > ---
> > >
> > > **Q6: I'm not sure if my understanding is correct: Although the WDT and IWDT proposed in this paper are based on existing wavelets, they constitute a different set of functions from the perspective of the "mother wavelet." Furthermore, WDT and IWDT satisfy the necessary conditions for wavelets and are capable of extracting derivative features.**
> > >
> > > **A6:**
> > > You are entirely correct. From the theoretical perspective of the "mother wavelet," WDT and iWDT indeed operate on a distinct set of derivative basis functions ($\psi^{(n)}$) that satisfy the necessary admissibility conditions (e.g., energy conservation proven in Theorem B.2). While our practical implementation utilizes the "standard WT + rescaling" strategy for computational efficiency (as detailed in **A5**), this operation is mathematically equivale to projecting the time series onto this new set of derivative bases. Consequently, these functions are uniquely capable of explicitly extracting derivative features (rate of change), thereby exposing non-stationary dynamics that standard wavelets might overlook.
> > >
> > > ---
> > >
> > > **Q7: The paper also has some limitations. For instance, in my view, designing a basis that satisfies numerous stringent properties is not straightforward; therefore, it would be preferable to provide more intuitive explanations of what these properties signify (though this may indeed be challenging).**
> > >
> > > **A7:**
> > > We agree that designing new bases satisfying these stringent properties is indeed challenging. We will add a discussion in the revised manuscript to intuitively explain the significance of these properties (e.g., Energy Conservation in Theorem B.2) and provide an outlook on potential future basis designs.
> > >
> > > ---
> > >
> > > **Q8: Additionally, the paper would benefit from demonstrating its applicability to wavelet bases other than db1, as the db1 (Haar) basis appears somewhat too simplistic.**
> > >
> > > **A8:**
> > > We have compared the performance of WaveTS using different wavelet bases (e.g., bior1.1, rbiol1.1) in Appendix D.8, Table 18 of the current paper. We apologize for not referencing this clearly in the main text and will explicitly highlight this comparison in the revised version.
> > >
> > > ---
> > >
> > > **Q9: I hope the authors could elaborate further on these points in the manuscript—additional discussion would likely enhance the paper’s clarity and impact. I appreciate the authors’ thoughtful reply, and I will raise my score accordingly.**
> > >
> > > **A9:**
> > > We sincerely thank you for your valuable suggestions, which are crucial for enhancing our paper's quality, and for your decision to raise the score. We will faithfully implement all promised revisions. To avoid confusion while other reviewers are assessing and reviewing the current version, we plan to finalize these changes and upload the ultimate revised manuscript before the rebuttal period concludes. Thank you again for your constructive feedback.

---

### Official Review · Reviewer_3ZMM · 2025-10-25

**Soundness:** 3
**Presentation:** 3
**Contribution:** 3
**Rating:** 6
**Confidence:** 4

**Summary:**

This paper introduces the **Wavelet Derivative Transform (WDT)** for deep time series forecasting, addressing the limitations of Fourier-based and standard wavelet transforms. WDT applies wavelet analysis to the derivative of a time series, improving sensitivity to abrupt changes while preserving trends. The authors propose **WaveTS**, a multi-branch architecture that processes different derivative orders, refines coefficients with Frequency Refinement Units, and reconstructs sequences using the inverse WDT. On several time series benchmarks, WaveTS outperforms state-of-the-art models.

**Strengths:**

1. The paper is well-structured, clearly written, and easy to follow.

2. The core innovations are well-expressed and make sense.

3. The experiments are thorough, and the results surpass state-of-the-art methods.

**Weaknesses:**

1. The layout of Figure 2 appears cluttered, with too much text and dense labeling, making it hard to interpret. A revision is recommended.

2. Line 302 mentions the benefits of adding backcast loss, but there is no ablation study to support this. It should be added.

3. The citation format in Table 3 is incorrect; `\cite` should be replaced with `\citep`.

**Questions:**

See **Weakness.**

---

> ### Author Response · Authors · 2025-11-21
>
> ## **Response to Reviewer 3ZMM**
>
> We sincerely thank Reviewer 3ZMM for the positive and constructive feedback on our work. Below we address each of your concerns point by point, and we have updated the revised manuscript accordingly.
>
> ----
>
> **Q1: The layout of Figure 2 appears cluttered, with too much text and dense labeling, making it hard to interpret. A revision is recommended.**
>
> **A1:** We agree that the previous version of Figure 2 was text-heavy and could be visually overwhelming. We also note that Reviewer v31u raised a similar concern about the readability of this figure. In the revised manuscript, we have redesigned Figure 2 to improve clarity and aesthetics:
> - We simplified the visual layout and reduced the amount of text directly embedded in the figure.
> - We adjusted the fonts, spacing, and grouping of components to make the architecture easier to read at a glance.
> We kindly invite you to re-download the updated PDF and refer to the new Figure 2 to see these improvements.
>
> ---
>
> **Q2: Line 302 mentions the benefits of adding backcast loss, but there is no ablation study to support this. It should be added.**
>
> **A2:** In Appendix D.3, we discussed the effect of backcast loss
> of the original submission. We apologize for not explicitly highlighting the ablation study on the backcast loss in the main text of the original submission. In particular, Table 10 reports an ablation study on the four ETT datasets, where we compare a version of WaveTS trained with forecast loss only (F), and a version trained with both backcast + forecast loss (BF). As shown in that table, adding the backcast loss consistently improves forecasting accuracy on all four ETT benchmarks, supporting our claim in Line 318 (revised version).
>
> To further strengthen this point and respond to your suggestion, we perform the ablation study on the remaining time series datasets used in Appendix D.1
> (i.e., Traffic, Electricity, Weather, and Exchange). The new results are reported in tables below, where (F) means only forecasting, (BF) means both backcasting and forecasting.
> As can be seen, across all these datasets and horizons, the (BF) variant consistently outperforms the (F) variant in both MSE and MAE, confirming that incorporating the backcast loss is beneficial for training WaveTS.
>
> **Backcast loss ablation across four forecasting horizons (MSE):**
> |**Traffic**|WaveTS(F)|WaveTS(BF)|**Electricity**|WaveTS(F)|WaveTS(BF)|**Weather**|WaveTS(F)|WaveTS(BF)|**Exchange**|WaveTS(F)|WaveTS(BF)|
> |:-:|:-:|:-:|:-:|:-:|:-:|:-:|:-:|:-:|:-:|:-:|:-:|
> |96|0.386|**0.382**|96| 0.138 |**0.131**|96|0.169 |**0.167**|96 |0.090|**0.086**|
> |192|0.400|**0.394**|192|0.148|**0.146**|192|0.220|**0.210**|192|0.184|**0.177**|
> |336|0.412|**0.409**|336|0.165|**0.162**|336|0.261|**0.256**|336|0.327|**0.322**|
> |720|0.460|**0.447**|720|0.210|**0.200**|720|0.320|**0.315**|720|0.892|**0.860**|
>
> **Backcast loss ablation across four forecasting horizons (MAE):**
> |**Traffic**|WaveTS(F)|WaveTS(BF)|**Electricity**|WaveTS(F)|WaveTS(BF)|**Weather**|WaveTS(F)|WaveTS(BF)|**Exchange**|WaveTS(F)|WaveTS(BF)|
> |:-:|:-:|:-:|:-:|:-:|:-:|:-:|:-:|:-:|:-:|:-:|:-:|
> |96|0.270|**0.266**|96|0.231|**0.227**|96|0.236|**0.223**|96|0.214|**0.204**|
> |192|0.278|**0.270**|192|0.246|**0.240**|192|0.270|**0.258**|192|0.310|**0.300**|
> |336|0.280|**0.278**|336|0.260|**0.256**|336|0.300|**0.294**|336|0.427|**0.411**|
> |720|0.302|**0.298**|720|0.292|**0.288**|720|0.344|**0.336**|720|0.720|**0.693**|
>
> ---
>
> **Q3: The citation format in Table 3 is incorrect; `\cite` should be replaced with `\citep`.**
>
> **A3:** Thank you for this careful observation. In the revised manuscript, we have corrected all citations in Table 3 from `\cite{·}` to `\citep{·}`, which aligns with the required citation style. You can verify this change in the updated PDF.
>
> ---
>
> **We again thank Reviewer 3ZMM for the careful reading and insightful review of our paper. Should you have any further concerns or suggestions, we would be very happy to address them.**

---

> > ### Comment · Reviewer_3ZMM · 2025-11-26
> >
> > Thanks for the response. I will keep my rating.

---

> > > ### Author Response · Authors · 2025-11-26
> > >
> > > Thank you for your response and for maintaining your positive evaluation. We greatly appreciate your thorough review and constructive feedback throughout this process. Should you have any additional questions or concerns, we remain available for further discussion. Thank you once again for your time.

---

### Official Review · Reviewer_v31u · 2025-10-29

**Soundness:** 3
**Presentation:** 3
**Contribution:** 3
**Rating:** 6
**Confidence:** 3

**Summary:**

This paper focuses on the task of time series forecasting. The existing methods still struggle in capturing multi-scale,  time-sensitive patterns. To solve these challenges, the proposed method uses multi-order Wavelet Derivative Transform (WDT), supported by theoretical analysis.

**Strengths:**

S1: The method is supported by theoretical analysis.

S2: The experiment is extensive.

S3: The idea of using multi-order Wavelet Derivative Transform is novel.

**Weaknesses:**

W1: The explanation of Figure 1, which is central to the paper's motivation, requires further clarification. The text states that ''(e)(f)the Fourier-derivative makes the spectrum stationary yet discards macro-trend information''. However, it is not immediately clear from the visual evidence provided in the figure how this ''discarding'' of the macro-trend is demonstrated. The similar problem exists for the sentence of  ''(e)(g)(h) The Wavelet Derivative Transform (WDT) retains those trends while offering complementary detail.'' It's hard to figure out the subfig (e)(f)(g)(h), which makes the motivation confused.

W2: Lack of clarity on the method's domain of superiority and failure modes. The paper provides a comprehensive evaluation of WaveTS on standard benchmarks but lacks a clear delineation of the specific time series characteristics for which the proposed method is most and least suited. Given the non-trivial complexity introduced by the multi-branch WDT architecture, it is crucial to understand its performance boundaries.

W3: The figure 2, i.e., the framework of WaveTS, is too complex to understand.

W4: Some typos. According to LIne 322, Electricity is used. Is this called ECL in Table 1?

**Questions:**

Please see the weaknesses above.

---

> ### Author Response · Authors · 2025-11-21
>
> ## **Response to Reviewer v31u**
> We thank Reviewer v31u for the constructive feedback. Below we address each of your concerns point by point. We have revised the corresponding sections in the manuscript accordingly.
>
> ---
> **Q1: The explanation of Figure 1 requires further clarification.**
>
> **A1:**
> In the revised manuscript, we have clarified the caption of Figure 1, explicitly describing the roles of subfigures (e)--(h). In particular, we now state that in (f) the Fourier-derived series mainly contains high-frequency fluctuations around zero and thus fails to reflect the global trend in (e), whereas in (g)--(h) WDT simultaneously captures the low-frequency trend and high-frequency fluctuations. These clarifications are highlighted in red in the updated PDF. We kindly invite the reviewer to download the revised version and check the marked portions (in red) of the Figure 1 caption.
>
> ---
> **Q2: Lack of clarity on the method's domain of superiority and failure modes.**
>
> **A2:**
> To obtain a clearer picture, we additionally construct three synthetic multivariate datasets of length $(T=12000)$ with $(D=7)$ variables, each component following: $X _ d(t)=(1-\lambda) S _ d(t)+\lambda T _ d(t)+\varepsilon _ d(t),$
> where $S _ d(t)=\sum _ {k=1}^{K} a _ {d,k} \sin(2\pi f _ k t+\phi _ {d,k})$
> is a seasonal component composed of multiple sinusoids with distinct frequencies $f _ k$, amplitudes $a _ {d,k}$, and phases $\phi _ {d,k}$; $T _ d(t)$ is a piecewise-linear trend function whose slope changes at several pre-specified change points to induce non-stationary behavior; and $\varepsilon _ d(t)\sim\mathcal{N}(0,\sigma^2)$ is zero-mean Gaussian noise.
>
> We set $\lambda \in 0.25, 0.5, 0.75$ to synthesize three datasets: $\lambda=0.25$ for a "Strong-Seasonal-Weak-Trend" dataset, $\lambda=0.5$ for a "Balanced-Trend-Balanced-Seasonal" dataset, and $\lambda=0.75$ for a "Strong-Trend-Weak-Seasonal" dataset. We compare WaveTS with several mainstream SOTA models: FITS, WPMixer and PatchTST. The input length is fixed to 512. The results are summarized in tables below.
>
> **Strong-Seasonal-Weak-Trend**
> | |WaveTS||FITS||WPMixer||PatchTST||
> |:-:|:-:|:-:|:-:|:-:|:-:|:-:|:-:|:-:|
> ||MSE|MAE|MSE|MAE|MSE|MAE|MSE|MAE|
> |96|0.065|0.204|0.162|0.321|**0.064**|**0.201**|0.070|0.211|
> |192|0.065|0.203|0.323|0.461|**0.063**|**0.201**|0.070|0.210|
> |336|0.064|0.203|0.325|0.465|**0.063**|**0.201**|0.070|0.210|
> |720|0.066|**0.204**|0.296|0.439|**0.065**|**0.204**|0.071|0.213|
>
> **Balanced-Trend-Balanced-Seasonal**
> | |WaveTS||FITS||WPMixer||PatchTST||
> |:-:|:-:|:-:|:-:|:-:|:-:|:-:|:-:|:-:|
> ||MSE|MAE|MSE|MAE|MSE|MAE|MSE|MAE|
> |96|0.085|0.232|0.151|0.306|**0.084**|**0.230**|0.088|0.235|
> |192|0.085|0.231|0.249|0.396|**0.084**|**0.230**|0.087|0.235|
> |336|0.085|0.231|0.260|0.407|**0.084**|**0.230**|0.088|0.235|
> |720|**0.087**|**0.234**|0.265|0.408|0.088|0.235|0.093|0.241|
>
> **Strong-Trend-Weak-Seasonal**
> | |WaveTS||FITS||WPMixer||PatchTST||
> |:-:|:-:|:-:|:-:|:-:|:-:|:-:|:-:|:-:|
> ||MSE|MAE|MSE|MAE|MSE|MAE|MSE|MAE|
> |96|**0.088**|**0.234**|0.116|0.270|0.089|0.235|0.089|0.235|
> |192|**0.087**|**0.234**|0.145|0.303|0.088|0.235|0.089|0.236|
> |336|**0.088**|**0.235**|0.167|0.327|0.089|0.236|0.091|0.238|
> |720|**0.093**|**0.241**|0.212|0.368|**0.093**|**0.241**|0.107|0.260|
>
> We observe that WPMixer achieves the lowest MSE and MAE on the Strong-Seasonal-Weak-Trend dataset, with WaveTS delivering comparable performance, indicating that wavelet-based models are particularly effective when seasonal (periodic) patterns dominate. In these settings, they consistently outperform FITS and PatchTST. As the trend component becomes more pronounced in the Balanced-Trend-Balanced-Seasonal and Strong-Trend-Weak-Seasonal datasets, the errors of WaveTS increase compared with those in the Strong-Seasonal setting. Nevertheless, WaveTS attains the best performance on the Strong-Trend dataset, likely due to the effect of WDT in capturing macro-level trend patterns.
>
> Although WDT adopts a multi-branch design, the overall computational cost remains low. As shown in Table 3, WaveTS attains the lowest memory footprint and the fastest inference speed among all baselines, indicating that in practice its complexity is modest rather than prohibitive.
>
> ---
> **Q3: The figure 2, i.e., the framework of WaveTS, is too complex to understand.**
>
> **A3:**
> In the revised manuscript, we have redesigned Figure 2 to improve clarity and visual organization: we simplified the overall layout and reduced dense text inside the figure, and also adjusted fonts, spacing, and grouping of modules. We kindly invite you to re-download the updated PDF and refer to the new Figure 2 to see these improvements.
>
> ---
> **Q4: Some typos. According to LIne 322, Electricity is used. Is this called ECL in Table 1?**
>
> **A4:**
> Yes, ECL is the abbreviation we use for the Electricity dataset. As revision, we have clarified this in the text (Line 373) and added an explicit note near the first mention at Line 338 (in red) to avoid confusion.

---

> > ### Comment · Reviewer_v31u · 2025-11-26
> >
> > Thanks for the response. My main concerns have been addressed.

---

> > > ### Author Response · Authors · 2025-11-26
> > >
> > > Thank you for your response. We are very pleased to hear that your main concerns have been addressed. We truly appreciate your careful review and constructive comments.

---

### Official Review · Reviewer_W9At · 2025-10-31

**Soundness:** 3
**Presentation:** 3
**Contribution:** 3
**Rating:** 4
**Confidence:** 4

**Summary:**

This paper proposes a new frequency-domain forecasting architecture grounded in multi-order wavelet derivatives, achieving solid empirical gains and offering an interpretable multi-scale design. The work effectively bridges traditional wavelet analysis with modern deep learning frameworks.

**Strengths:**

1. The paper introduces a new idea, i.e., using multi-order wavelet derivatives, that extends traditional wavelet transforms to a learnable, multi-resolution representation explicitly tied to derivative order, bridging signal processing and deep learning perspectives.

2. The model achieves consistent performance gains across diverse benchmarks, demonstrating robustness to both long-term and short-term forecasting scenarios.

**Weaknesses:**

1. My main concern is that the paper appears conceptually similar to DeRiTS, with the primary difference being the replacement of the Fourier-based derivative operator by a wavelet-based one. The overall framework and motivation seem closely aligned, which raises questions about the degree of novelty.

2. The ablation section does not sufficiently isolate the contributions of wavelet decomposition, derivative order, and fusion strategy. It is unclear which component contributes most to the gains.

**Questions:**

1. Are the wavelet parameters (e.g., basis, scale) fixed or learnable during training?

2. Could the proposed wavelet derivative operator be integrated into other architectures, such as CNNs or MLP-Mixers, beyond the current framework?

---

> ### Author Response · Authors · 2025-11-21
> **Response to Weakness 1**
>
> ## **Response to Reviewer W9At**
>
> We sincerely appreciate Reviewer W9At for the thoughtful feedback. We address the concerns below and will address any remaining issues in the subsequent rebuttal. We also kindly invite the reviewer to download the revised PDF, in which several modifications have been included.
>
> ---
>
> **Q1: My main concern is that the paper appears conceptually similar to DeRiTS, with the primary difference being the replacement of the Fourier-based derivative operator by a wavelet-based one. The overall framework and motivation seem closely aligned, which raises questions about the degree of novelty.**
>
> **A1:**
> There might be some misunderstanding regarding the similarity between DeRiTS and our proposed WaveTS. WaveTS is not merely a "wavelet version of DeRiTS", but differs fundamentally in its **motivation**, **architecture**, **training objective**, and other aspects, as summarized below.
>
> **1. Different motivations: Stationarity only vs macro-micro decomposition**
>
> - DeRiTS motivates Fourier Derivative Transform (FDT) mainly to obtain more stationary frequency representations (Lines 17-21 in their Abstract). In contrast, the Introduction and Figure 1 of our paper argue that derivative-only frequency representations tend to suppress macro-trend information.
>
> - WaveTS is designed to jointly capture macro trends and micro variations across various scales using WDT, explicitly targeting both stationary and non-stationary patterns in the time-frequency domain, rather than only improving stationarity (from Figure 1 in our paper, WDT captures complementary non-stationary patterns in (g) that are not present in FDT’s representation in (f).)
>
> **2. Different architectural and technical designs**
>
> - WaveTS uses instance normalization (Section 3.2.1) to mitigate distribution shifts and enable more accurate forecasting, which DeRiTS does not.
>
> - DeRiTS employs an Order-adaptive Fourier Convolution in their Section 4.3 that filters part of the spectrum, downsampling in frequency. In contrast, WaveTS uses a Frequency Refinement Unit that operates on all WDT coefficients and expands their length by interpolation (Eq. (4)), implementing frequency-domain super-resolution, performing frequency upsampling instead of downsampling. Such frequency upsampling avoids information loss brought by frequency downsampling while capturing fine-grained patterns.
>
> - WaveTS specifies an explicit temporal fusion scheme (Branch Aggregation) in Section 3.2.2: After iWDT on each branch, we concatenate along the time dimension and apply a linear projection as in Eq. (6), and we further ablate temporal vs variate fusion in Appendix D.4. However, the exact fusion mechanism is not described with equal clarity in DeRiTS. Moreover, as shown in our experiments in Table 11 (Appendix D.4), DeRiTS underperforms WaveTS under both fusion strategies, further underscoring the superiority of WaveTS.
>
> **3. Different training objectives for frequency-domain models**
>
> - DeRiTS optimizes a standard L2 forecasting loss only on future horizons.
>
> - WaveTS proposes a backcasting while forecasting objective (Eq. (7), Section 3.2.3), which simultaneously reconstructs the past and predicts the future in the frequency domain. This change of supervision paradigm is tailored to Frequency-domain Representation Learning and leads to consistent gains. We provide an ablation study on this strategy in Table 10 (Appendix D.3). This ablation was included in the original submission, but we apologize for the oversight of not explicitly mentioning it in the main text.
>
> **Empirical comparison and reimplementation of DeRiTS**
> By carefully examining the main experiments of DeRiTS (Table 1), we observe that the reported MAE values are numerically far from widely accepted results on the same benchmarks. The reviewer can directly compare these numbers with those in recent peer-reviewed works (e.g. Table 1 in [1] and [2]), where the gap is clearly of a different order of magnitude). Since the code of DeRiTS is not  released, we cannot verify its performance as reported in their paper. To ensure a fair and reproducible comparison, we therefore re-implemented DeRiTS under our standardized experimental protocol and report the results in our paper. We are happy to provide our reproduced code and checkpoints. More detailed empirical comparisons between DeRiTS and WaveTS can be found in Tables 1-3, 6, 8-11 and Figures 11-12 of our paper.
>
> **Taken together, WaveTS is not conceptually equivalent to DeRiTS.**
>
> **Reference:**
> * [1] NeurIPS'25 Multi-Modal View Enhanced Large Vision Models for Long-Term Time Series Forecasting.
> * [2] NeurIPS'25 Enhancing the Maximum Effective Window for Long-Term Time Series Forecasting.

---

> ### Author Response · Authors · 2025-11-21
> **Response to Weakness 2, Question 1 and Question 2**
>
> **Q2: The ablation section does not sufficiently isolate the contributions of wavelet decomposition, derivative order, and fusion strategy. It is unclear which component contributes most to the gains.**
>
> **A2:** Thank you for pointing this out. We address your concerns point by point below.
>
> **1. Ablation on Fusion Strategy**
>
> The fusion strategy is ablated in Appendix D.4 (Table 11). We apologize for not clearly mentioning the ablations in the main text. In Appendix D.4, we compare temporal-dimension fusion with variate-dimension fusion under identical settings and observe that temporal fusion consistently performs better.
>
> **2. Ablation on Wavelet Decomposition**
>
> We have already provided ablations on the "wavelet decomposition" in Appendix D.9 (Table 19). By replacing WDT with Discrete Fourier Transform, we obtain the variant FourierTS (Figure 10), whose forecasting performance is consistently worse than WaveTS on all 8 datasets (Table 19).
>
> **3. Ablation on Different Derivative Orders**
>
> We have examined how different derivative orders affect forecasting performance in Figure 3, Figure 12, and Table 12 (Appendix D.5). We further investigate "Which orders suit which datasets?" in Appendix D.7, where the numerical results provide clearer insights into the impact of derivative orders in WaveTS.
>
> ---
>
> **Q3: Are the wavelet parameters (e.g., basis, scale) fixed or learnable during training?**
>
> **A3:** In our current implementation, both the wavelet basis and the scale configuration are predefined hyperparameters and kept fixed during training, they are not learnable. Concretely, as stated in Line 355 (revised version), we uniformly use the db1 basis across all experiments. However, notice that our framework is also compatible with other wavelet bases. As for the choice of wavelet basis, Appendix D.8 provides sensitivity study over different mother wavelets, showing that WaveTS remains effective and robust under alternative bases. Similarly, for the scale (number and range of decomposition levels) in WDT, we systematically analyze its impact in Appendix D.5 (Table 13) as well as Figures 3 and 9, where we vary the scale configuration and report the resulting forecasting performance. These experiments demonstrate that our method is reasonably robust to scale choices and help justify the default configuration used in the main results.
>
> ---
>
> **Q4: Could the proposed wavelet derivative operator be integrated into other architectures, such as CNNs or MLP-Mixers, beyond the current framework?**
>
> **A4:** Conceptually, our proposed WDT is architecture-agnostic and can be combined with different representation learners. Motivated by your suggestion, we replace the Frequency Refinement Unit (FRU) in Eq. (4) with CNN and MLP-Mixer blocks, respectively. Concretely, instead of Eq. (4), we introduce
> two variants: (i) WaveTS(CNN), which maps the WDT coefficients through $\widehat{LL} _ {K}^{(n)} = \mathrm{CNN}\bigl(LL _ {K}^{(n)}\bigr), \widehat{\mathbf{LH}} _ {K}^{(n)} = \mathrm{CNNs}\bigl(\mathbf{LH}_{K}^{(n)}\bigr)$; (ii) WaveTS(MLP-Mixer), which maps the WDT coefficients through $\widehat{LL} _ {K}^{(n)} =\mathrm{MLPMixer}\bigl(LL _ {K}^{(n)}\bigr),\widehat{\mathbf{LH}} _ {K}^{(n)} = \mathrm{MLPMixers}\bigl(\mathbf{LH} _ {K}^{(n)}\bigr)$.
> The MSE results are shown in the tables. In all cases, both WaveTS(CNN) and WaveTS(MLP-Mixer) perform worse than the original WaveTS, even though FRU in our design is just a single-layer MLP and is the only trainable component on top of WDT. This is likely because the FRU’s simple structure is capable of modeling WDT coefficients effectively, while CNNs and MLP-Mixers impose inductive biases (e.g., spatial locality or token mixing) that are less compatible with the derivative wavelet domain. We thank the reviewer for this valuable suggestion, which motivates us to explore other architectures that are more specifically tailored to WDT coefficients in future work.
>
>
> |**ETTh1**|WaveTS|WaveTS(CNN)|WaveTS(MLP-Mixer)|**ETTh2**|WaveTS|WaveTS(CNN)|WaveTS(MLP-Mixer)|
> |:-:|:-:|:-:|:-:|:-:|:-:|:-:|:-:|
> |96|**0.367**|0.384|0.370|96|**0.267**|0.296|0.273|
> |192|**0.404**|0.418|0.435|192|**0.332**|0.354|0.337|
> |336|**0.427**|0.430|0.448|336|**0.349**|0.377|0.359|
> |720|**0.440**|0.478|0.475|720|**0.380**|0.420|0.397|
>
> |**Exchange**|WaveTS|WaveTS(CNN)|WaveTS(MLP-Mixer)|**Weather**|WaveTS|WaveTS(CNN)|WaveTS(MLP-Mixer)|
> |:-:|:-:|:-:|:-:|:-:|:-:|:-:|:-:|
> |96|**0.086**|0.681|0.095|96|**0.167**|0.186|0.170|
> |192|**0.177**|0.841|0.194|192|**0.210**|0.222|0.216|
> |336|**0.322**|1.078|0.358|336|**0.256**|0.262|0.270|
> |720|**0.860**|1.866|1.277|720|**0.315**|0.320|0.333|

---

> ### Author Response · Authors · 2025-11-27
> **Follow-up to Reviewer W9At**
>
> We sincerely thank Reviewer W9At again for the detailed and constructive feedback. In our earlier responses, we have clarified the differences between WaveTS and DeRiTS in terms of motivation, architecture, and training objectives, and we have also expanded the ablation studies on fusion strategy, wavelet decomposition, and derivative orders, as well as the clarification of wavelet parameters and the integration of WDT into CNN/MLP-Mixer variants.
>
> As the rebuttal phase is approaching its end, we hope that our revisions and additional analyses have helped address your main concerns about novelty and empirical validation. If there remain any aspects that you feel are not yet sufficiently clarified, we would be very happy to provide further explanations or additional small-scale analyses within the remaining rebuttal timeline.
>
> Thank you once again for your time, efforts, and thoughtful comments on our work.

---

### Author Response · Authors · 2025-11-30
**TL;DR: Summary of Reviews, Scores, and Author Rebuttals**

Dear new AC,

**Thank you for your time and dedication, not only to our paper but also to the broader ICLR community, especially under the current unusual and challenging circumstances.** All four reviewers provided detailed initial reviews, and three of them further participated in the rebuttal discussion. Before the rollback of scores to their pre-discussion state:

- Reviewer 5xKq increased the rating from **4 to 8** after our detailed responses.
- Reviewers v31u and 3ZMM stated that their main concerns had been resolved and kept their scores at **6**.
- Reviewer W9At did not post a follow-up comment and remained at **4**.

**In addition, we have already incorporated several changes directly into the paper. In the revised PDF, the main updates are:**

* We rewrote the caption and surrounding text of Figure 1 to more clearly explain our research motivation (addressing Reviewer v31u Q1).
* We redesigned Figure 2 with a simpler layout and clearer module grouping, and updated its description (addressing Reviewer v31u Q3 and Reviewer 3ZMM Q1).
* We explicitly stated that ECL denotes the Electricity dataset (addressing Reviewer v31u Q4).
* We corrected the citation style in Table 3 (using `\citep{·}`) (addressing Reviewer 3ZMM Q3).
* In the Introduction, we added a concise summary of the WDT/iWDT and clarified why they are non-trivial and central to WaveTS (addressing Reviewer 5xKq Q1).

**Below, we briefly summarize how we addressed the key points raised by each reviewer:**

* **Reviewer 5xKq (score raised from 4 to 8 on 25 Nov)**

  * **Q1.** We revised the Introduction to make clear that WDT is a new, invertible, energy-preserving transform with a derivative interpretation, and explained why this is non-trivial and central to WaveTS.

  * **Q2.** We explained that direct differentiation of discrete sequences is unstable, and that WDT/iWDT instead operate in the wavelet domain via "orthonormal DWT + scale-wise rescaling", yielding a stable, invertible mapping.

  * **Q3.** We clarified that WDT is built by rescaling coefficients of a standard DWT basis with perfect reconstruction, so iWDT naturally ensures a lossless, energy-stable representation that can be mapped back to the time domain.

  * **Q4.** We discussed standard wavelet requirements and pointed to our energy conservation result for WDT, contrasting it with methods that discard parts of the spectrum and thus lose information.

  * **Q5.** We stressed that we do not numerically construct $\psi^{(n)}$; WDT is implemented as "one standard DWT + scale-wise rescaling", with db1’s high-pass filter acting as a discrete difference operator, giving a computationally efficient and theoretically justified implementation that generalizes to other bases (Appendix D.8).

* **Reviewer v31u (kept the score of 6, with main concerns resolved on 26 Nov)**

  * **Q1.** We rewrote the caption and text so that it is explicit that Fourier derivatives suppress global trends, while WDT retains trends and captures fine-scale variations.

  * **Q2.** We added a synthetic study showing that WaveTS is particularly effective under strong seasonality and non-stationary trends, and we identified regimes where other models can be competitive.

  * **Q3.** We simplified Figure 2 and highlighted that WaveTS has the lowest memory usage and fastest inference among baselines, indicating that the multi-branch design is practically lightweight.

  * **Q4.** We clarified that ECL denotes the Electricity dataset and unified the terminology between text and tables.

* **Reviewer 3ZMM (kept the score of 6, explicitly stated on 26 Nov)**

  * **Q1.** We redesigned Figure 2 with less crowded labels and clearer module grouping, making the architecture easier to follow.

  * **Q2.** We emphasized the original ETT ablation and added new experiments on additional datasets, consistently showing that adding a backcast term improves both MSE and MAE.

  * **Q3.** We corrected the citation style in Table 3 and related LaTeX formatting, as requested.

* **Reviewer W9At (with an initial score of 4 and no follow-up as of 29 Nov)**

  * **Q1.** We clarified that WaveTS differs from DeRiTS in motivation, architecture, and training objective, and our re-implementation of DeRiTS under a unified protocol shows consistent improvements by WaveTS.

  * **Q2.** We highlighted existing ablations that examine the effects of the fusion scheme, WDT vs. DFT (FourierTS), and derivative orders/scales, helping clarify which design choices contribute most to the gains.

  * **Q3.** We stated that wavelet basis and scales are fixed hyperparameters, and sensitivity studies indicate stable performance under alternative bases and scale settings.

  * **Q4.** We conducted additional experiments showing that replacing the FRU with CNN or MLP-Mixer blocks is feasible, indicating that WDT is architecture-agnostic, while the simple FRU still performs best in our experiments.

Sincerely,

The Authors of Paper 12589

---

### Meta-Review · Area_Chair_VTM9 · 2026-01-19

**Summary:**

The paper proposes a new technique (Wavelet Derivative Transform) that incorporates the wavelet transform into a multi-brach architecture that "decomposes the input series into multi-scale time-frequency coefficients, refines them via linear layers, and reconstructs them into the time domain via the inverse WDT".

The main weaknesses identified by the reviewers are as follows:

W1. Limitations in terms of novelty and differentiation with respect to prior work.

W2. The superiority of the method was not clearly demonstrated, and the cases when the method is expected to do well vs not do well are not clear.

W3. Ablation studies not sufficiently clear.

W4. Poor writing that impedes the understanding of the method, with proofs seemingly disconnected from the broader discussion.

**Reviewer Concerns:**

W1. The authors convincingly argued why their method is different from DeRiTS. There are other methods out there that focus on frequency-based decomposition, however, as the reviewers did not reference them, the authors have no way of specifically explain the differences. It seems that the other methods, while not exactly the same, exploit similar characteristics in data.

W2. Despite the authors' answer, this weakness is still not sufficiently addressed, as the gains in performance are tiny, even in the case when the method is supposed to do well (Strong-Trend-Weak-Seasonal). The one aspect the method that arguably makes it better than other methods is its efficiency.

W3. The authors have provided a response which I consider to be adequate.

W4. Reviewer 5xKq identified some issues with the writing, which the authors clarified, the reviewer raising their score from a 4 to an 8. However, if the writing was such an issue, there is the question of how the camera-ready will look like,

**Reviewer Scores:**

I do not know whether the reviewers would have actually changed their scores, however, reviewer stated their intent to raise their score from a 4 to an 8.

---

### Decision · Program_Chairs · 2026-01-26

Reject